# Histone-methyltransferase KMT2D deficiency impairs the Fanconi anemia/BRCA pathway upon glycolytic inhibition in squamous cell carcinoma

Wei Liu [1,2,4], Hongchao Cao[1,2,4], Jing Wang[1,2], Areeg Elmusrati[1,2], Bing Han[1,2], Wei Chen[1,2], Ping Zhou[1,2], Xiyao Li[1,2], Stephen Keysar[3], Antonio Jimeno[3] & Cun-Yu Wang [1,2] ✉

Histone lysine methyltransferase 2D (KMT2D) is the most frequently mutated epigenetic modifier in head and neck squamous cell carcinoma (HNSCC). However, the role of KMT2D in HNSCC tumorigenesis and whether its mutations confer any therapeutic vulnerabilities remain unknown. Here we show that KMT2D deficiency promotes HNSCC growth through increasing glycolysis. Additionally, KMT2D loss decreases the expression of Fanconi Anemia (FA)/BRCA pathway genes under glycolytic inhibition. Mechanistically, glycolytic inhibition facilitates the occupancy of KMT2D to the promoter/enhancer regions of FA genes. KMT2D loss reprograms the epigenomic landscapes of FA genes by transiting their promoter/enhancer states from active to inactive under glycolytic inhibition. Therefore, combining the glycolysis inhibitor 2-DG with DNA crosslinking agents or poly (ADP-ribose) polymerase (PARP) inhibitors preferentially inhibits tumor growth of KMT2D-deficient mouse HNSCC and patient-derived xenografts (PDXs) harboring KMT2D-inactivating mutations. These findings provide an epigenomic basis for developing targeted therapies for HNSCC patients with KMT2D-inactivating mutations.

HNSCC is a prevalent and aggressive malignancy that develops from the mucosal epithelium of the upper aerodigestive tract and frequently metastasizes to the cervical lymph nodes[1,2]. The current therapies available for HNSCC include surgery, radiotherapy, chemotherapy, and immunotherapy. However, the 5-year survival rate remains low due to the high incidence of treatment resistance, regional metastasis, and recurrence[3,4]. Growing evidence shows that specific gene mutations in cancer cells can confer druggable vulnerabilities and yield valuable therapeutic strategies[5–7]. Therefore, discovering vulnerabilities associated with frequent gene mutations in HNSCC can

help identify potential therapeutic targets and improve outcomes of HNSCC treatment.

KMT2D, also known as MLL4, is a member of the complex of proteins associated with Set1 (COMPASS)-like family and is responsible for catalyzing the mono-, di-, and tri-methylation at lysine 4 on histone H3 (H3K4)[8]. In addition to its methyltransferase activity, KMT2D is required for histone acetyltransferase CBP/P300-mediated acetylation at lysine 27 on histone H3 (H3K27ac)[9,10]. *KMT2D* has emerged as one of the most frequently mutated genes in various types of cancer[11,12]. According to The Cancer Genome Atlas (TCGA) database, KMT2D is

[1]Jonsson Comprehensive Cancer Center, University of California, Los Angeles, Los Angeles, CA, USA. [2]Laboratory of Molecular Signaling, Division of Oral and Systemic Health Sciences, School of Dentistry, University of California, Los Angeles, Los Angeles, CA, USA. [3]Division of Medical Oncology, Department of Medicine, University of Colorado Anschutz Medical Campus, Aurora, CO, USA. [4]These authors contributed equally: Wei Liu, Hongchao Cao. ✉e-mail: cwang@dentistry.ucla.edu

the most highly mutated epigenetic modifier in HNSCC, indicating its fundamental role in HNSCC tumorigenesis[13]. However, the role of KMT2D in HNSCC is still poorly understood as existing studies showed controversial effects[14].

Notably, several studies demonstrated that KMT2D deficiency has great therapeutic potential in various cancer types. For instance, three independent studies have recently revealed that KMT2D loss enhances the Warburg effect in lung cancer[15], melanoma[16], and pancreatic cancer[17]. The Warburg effect, also called aerobic glycolysis, is a fundamental metabolic alteration in cancer cells characterized by increased glucose uptake and lactate production. Although the Warburg effect facilitates tumor growth and survival by providing ATP and metabolic intermediates, it renders cancer cells sensitive to glycolytic inhibition[18]. Indeed, the pharmacological inhibition of glycolysis selectively impedes the growth of lung cancer and melanoma with KMT2D-inactivating mutations[15,16]. Moreover, a recent study showed that KMT2D loss renders lung squamous cell carcinoma (LUSC) therapeutically vulnerable to receptor tyrosine kinase (RTK)-RAS inhibition[19]. Therefore, the potential therapeutic vulnerabilities associated with KMT2D loss encourage us to explore more effective treatments for HNSCC patients with somatic KMT2D mutations.

In this study, we demonstrate that KMT2D deficiency activates oncogenic and metabolic signaling pathways, including glycolysis, mTORC1 signaling, and ribosome biogenesis, to promote the aggressive growth of HNSCC. We find that the glycolytic inhibitor 2-DG preferentially impedes the tumor growth of KMT2D-deficient HNSCC. Further scrutinizing the epigenomic changes in HNSCC following glycolysis inhibition, we unexpectedly discover that KMT2D deficiency reprograms the epigenetic landscapes of HNSCC and impairs the expression of FA/BRCA pathway genes under glucose deprivation. Consequently, the impaired FA/BRCA pathway in KMT2D-deficient HNSCC under glycolytic inhibition renders these tumors hypersensitive to the combined treatment of glycolytic inhibitors and DNA-damaging agents or PARP inhibitors. This finding provides an opportunity to develop a biomarker-driven therapeutic approach for patients with KMT2D deficient HNSCC.

## Results

### *Kmt2d* heterozygosity promotes HNSCC initiation, progression, and metastasis

To explore the role of KMT2D in HNSCC, we first examined the genetic alterations of KMT2D in various cancer types. TCGA database analysis by TIMER2.0 revealed that KMT2D is among the most frequently mutated genes in multiple cancer types. Notably, HNSCC, including human papillomavirus positive (HPV + ) and negative (HPV-) subsets, showed a relatively high mutation frequency of KMT2D (Supplementary Fig. 1a). Consistent with this finding, TCGA database analysis from the cBioPortal indicated that KMT2D genomic alterations occur in approximately 16% of HNSCC cases (Supplementary Fig. 1b). Further analysis of these *KMT2D* genetic mutations in HNSCC found that the majority of mutations were truncations (60.7%), which cause loss-of-function of *KMT2D* (Supplementary Fig. 1c). Moreover, TCGA database analysis showed that loss-of-function mutation of KMT2D was significantly associated with poorer overall survival (OS) in HNSCC (Supplementary Fig. 1d). To investigate the role of KMT2D in HNSCC tumorigenesis, we established *Kmt2d* heterozygous conditional knockout mouse (*K14^creER*;*Kmt2d^fl/+*: *Kmt2d*-HT) and then treated these mice together with wildtype mice (*K14^creER*;*Kmt2d^+/+*: *Kmt2d*-WT) with 4-nitroquinoline 1-oxide (4NQO) to induce SCC in the tongue. Of note, because the pups from homozygous mating are severely runted according to The Jackson Laboratory and our pilot studies, we utilized heterozygous *Kmt2d*-HT mice which grow normally in our studies. 4NQO-induced mouse HNSCC model could largely simulate human HNSCC initiation, progression, and lymph node metastasis[20] (Fig. 1a). Firstly, we confirmed the reduction of *Kmt2d* mRNA levels in the

tongue epithelia of *Kmt2d*-HT mouse by quantitative RT-PCR (qRT-PCR) (Supplementary Fig. 1e) after tamoxifen (Tam) treatment. After being treated with drinking water containing 4NQO for 16 weeks and then with normal water for another 10 weeks, *Kmt2d*-HT mice had increased HNSCC lesion areas compared with *Kmt2d*-WT mice (Fig. 1b). Decreased KMT2D protein levels in *Kmt2d*-HT mice tongue tumors were confirmed by immunohistochemistry (IHC) (Fig. 1c). Histological analysis found that the numbers, areas, and invasive grades of HNSCC were significantly increased in *Kmt2d*-HT mice compared with *Kmt2d*-WT mice (Fig. 1d, e). Consistent with aggressive HNSCC formation, tongue tumors in *Kmt2d*-HT mice showed higher levels of the cell proliferation marker Ki-67 (Fig. 1f). As lymph node metastasis is an important prognostic factor in HNSCC, we further checked the metastatic status of cervical lymph nodes which are the most common sites of HNSCC metastasis[3,21]. Both the areas and numbers of metastatic lymph nodes were significantly increased in *Kmt2d*-HT mice compared with *Kmt2d*-WT mice as determined by anti-pan-cytokeratin (PCK) immunostaining (Fig. 1g, h). These results suggest that *Kmt2d* loss promotes HNSCC initiation and development.

Next, we sought to investigate whether *Kmt2d* deficiency could accelerate HNSCC growth and metastasis after its initiation. As primary HNSCC in the tongue epithelia begins to form in 4NQO-treated mice at 20 weeks[20], we administered tamoxifen at 21 weeks to induce *Kmt2d* heterozygous deletion (Fig. 1i). Four weeks after *Kmt2d* heterozygous deletion, we collected the mouse tongues and cervical lymph nodes for further analysis. IHC staining confirmed the decreased protein levels of KMT2D in *Kmt2d*-HT mice (Supplementary Fig. 1f). We found that *Kmt2d*-HT mice showed significantly increased tumor lesion areas, SCC numbers, SCC areas, and tumor invasion grades as well as Ki-67 positive tumor cell numbers compared with *Kmt2d*-WT mice (Fig. 1j–l and Supplementary Fig. 1g), indicating that heterozygous deletion of *Kmt2d* could accelerate HNSCC progression. Furthermore, *Kmt2d* deficiency further promoted the cervical lymph node metastasis of HNSCC (Fig. 1m, n). Taken together, these results suggest that KMT2D functions as an important tumor suppressor in HNSCC and *Kmt2d* deficiency significantly promotes not only HNSCC initiation, but also HNSCC progression and metastasis.

### Loss of KMT2D promotes aggressive HNSCC growth via increasing glycolysis

To further investigate the role of KMT2D as a tumor suppressor in human HNSCC, we generated *KMT2D* knockout (KMT2D-KO) SCC23 HNSCC cells by CRISPR-Cas9 mediated gene editing. KMT2D mutations were confirmed by genomic sequencing (Supplementary Fig. 2a), and the knockout of KMT2D was subsequently confirmed by western blot (Fig. 2a). Loss of KMT2D markedly decreased H3K4me1 levels (Fig. 2a) but did not affect the expression of other *KMT2D* family genes (Supplementary Fig. 2b, c). SCC23 cells usually grow slowly and form small tumors when inoculated subcutaneously. However, loss of KMT2D significantly promoted subcutaneous tumor growth of KMT2D-KO SCC23 cells in nude mice compared with KMT2D-WT cells (Fig. 2b, c), although both cells showed similar rates of proliferation in vitro (Supplementary Fig. 2d). We also generated CRISPR-Cas9 mediated KMT2D-KO in SCC1 cells. The deletion of KMT2D in SCC1 cells was verified by genomic sequencing and western blot (Supplementary Fig. 2e, f). Consistent with the results in SCC23 cells, although the in vitro proliferation rates of SCC1 cells were not affected by the loss of KMT2D (Supplementary Fig. 2g), knockout of KMT2D significantly increased the subcutaneous tumor formation (Supplementary Fig. 2h).

To explore the mechanisms by which KMT2D loss promoted HNSCC, we performed RNA sequencing (RNA-seq) analysis of both KMT2D-KO and KMT2D-WT SCC23 cells. Gene set enrichment analysis (GSEA) showed that several crucial cancer hallmark signatures were enriched in KMT2D-KO SCC23 cells, including glycolysis, mTORC1

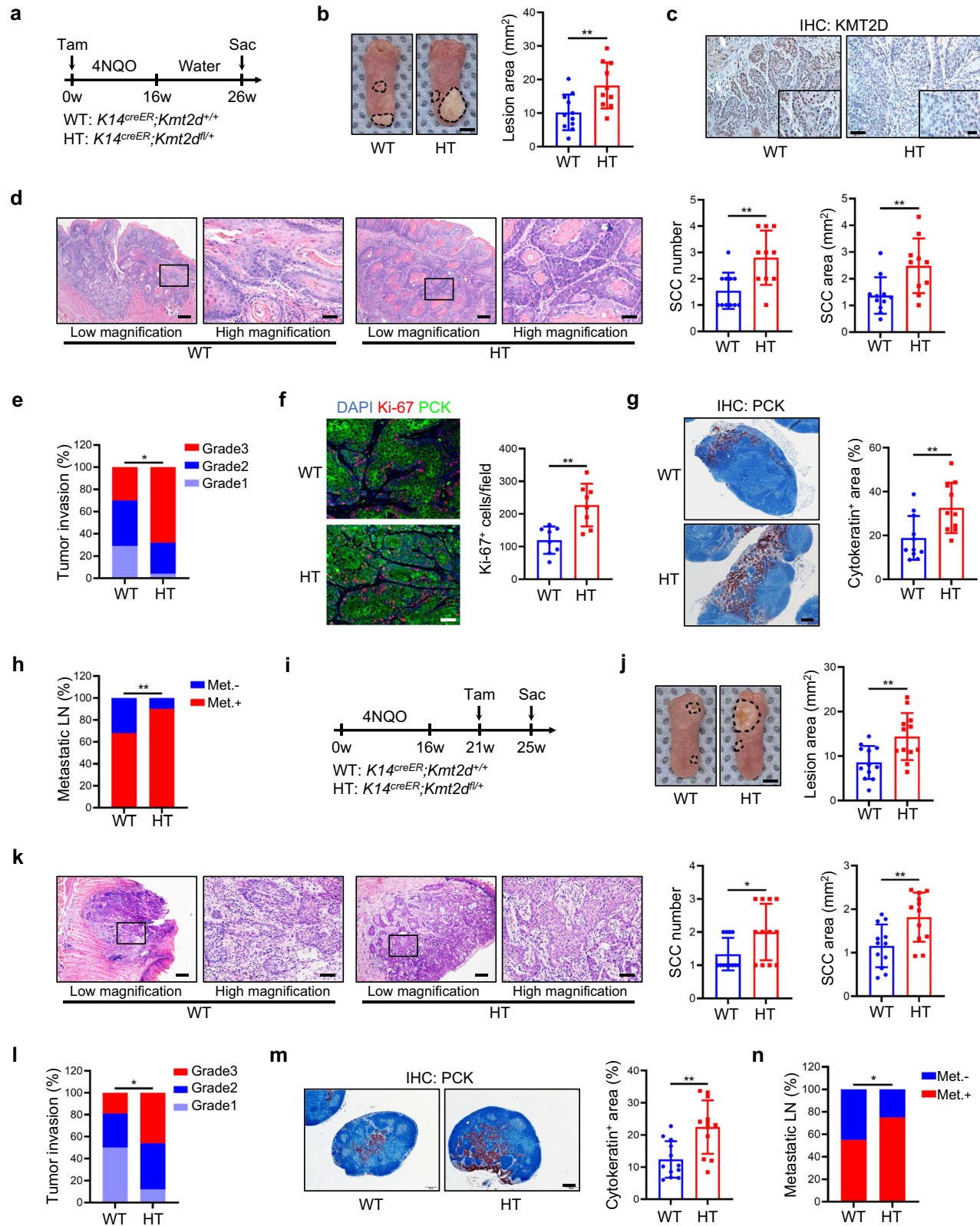

signaling, MYC targets, and ribosome biogenesis (Fig. 2d–g). Increased aerobic glycolysis, known as the Warburg effect, is the most fundamental metabolic alteration during malignant transformation[18]. Previously, KMT2D-mutant lung cancer, melanomas, and pancreatic cancer also showed enhanced glycolysis[15–17], indicating that hyperactive glycolysis is an important feature for KMT2D-mutant tumors among different types. In line with the GSEA data, transcriptomic

analysis by RNA-seq showed that the expression levels of glycolysis-related genes were significantly increased due to KMT2D loss (Fig. 2h). The upregulation of typical glycolytic genes such as *ENO1*, *GCLC*, *HK2*, *HKDC1*, *LDHB*, *PGK1*, and *SLC16A1* by KMT2D loss was confirmed by qRT-PCR (Fig. 2i), and the protein levels of LDHB and PGK1 were also confirmed by western blot (Fig. 2j). Additionally, glucose uptake and lactate release were substantially increased by KMT2D loss (Fig. 2k, l).

**Fig. 1 | Epithelial heterozygous deletion of *Kmt2d* promotes mouse HNSCC invasive growth and lymph node metastasis. a** Experimental design for 4NQO-induced primary mouse HNSCC. **b** Representative image of tongue lesions and quantification of lesion areas. Scale bar, 2 mm. Values are mean ± SD. *n* = 11:10. **p* < 0.01 by unpaired two-tailed Student's t test. **c** Representative IHC staining images of KMT2D in mouse HNSCC were selected from two independent experiments. *n* = 11:10. Enlarged images are in inserts. Scale bar, 50 μm. **d** Representative H&E staining of HNSCC and quantification of HNSCC number and area. Scale bar, 200 μm. Enlarged images are in the right panel. Scale bar, 50 μm. *n* = 11:10. ***p* < 0.01 by unpaired two-tailed Student's t test. **e** Quantification of HNSCC invasion grades. **p* < 0.05 by two-sided Cochran-Armitage test. **f** Representative IF staining and quantification of Ki-67 positive cells. Scale bar, 50 μm. Values are mean ± SD. *n* = 8:8. ***p* < 0.01 by unpaired two-tailed Student's t test. **g** IHC staining of PCK positive cells and quantification of the metastatic area in cervical lymph nodes. Scale bar,

200 μm. Values are mean ± SD. *n* = 11:10. ***p* < 0.01 by unpaired two-tailed Student's t test. **h** Quantification of metastatic lymph nodes. ***p* < 0.01 by two-sided chi-square test. **i** Experimental design for studying the effects of *Kmt2d* on the growth of 4NQO-induced mouse HNSCC. **j** Representative image of tongue lesions and quantification of lesion areas. Scale bar, 2 mm. Values are mean ± SD. *n* = 12:12. ***p* < 0.01 by unpaired two-tailed Student's t test. **k** Representative H&E staining of HNSCC and quantification of HNSCC number and area. Scale bar, 200 μm. Enlarged images are in the right panels. Scale bar, 50 μm. Values are mean ± SD. *n* = 12:12. **p* < 0.05, ***p* < 0.01 by unpaired two-tailed Student's t test. **l** Quantification of HNSCC invasion grades. **p* < 0.05 by two-sided Cochran-Armitage test. **m** IHC staining of PCK positive cells and quantification of metastatic area in lymph nodes. Scale bar, 200 μm. *n* = 12:12. ***p* < 0.01 by unpaired two-tailed Student's t test. **n** Quantification of metastatic lymph nodes. **p* < 0.05 by two-sided chi-square test. Source data are provided as a Source Data file.

The intracellular levels of glycolytic metabolites 2-phosphoglycerate (2-PG) and pyruvate were significantly higher in KMT2D-KO SCC23 cells than in KMT2D-WT cells (Fig. 2m, n). These results were consistent with the upregulated glycolytic genes in KMT2D-KO SCC23 cells. Moreover, *Kmt2d* deficiency increased the protein levels of LDHB and PGK1 in the 4NQO-induced mouse HNSCC as determined by IHC staining (Fig. 2o, p). Analysis of the TCGA database also showed a negative correlation between *KMT2D* mRNA levels and the mRNA levels of several glycolytic genes, including *ADH5*, *ENO1*, *GAPDH*, *LDHB*, *PDHA1*, and *TPI1* (Supplementary Fig. 2i).

Cancer cells with enhanced Warburg effect rely on glucose-dependent aerobic glycolysis for generating ATP and supporting metabolic function, thus making them susceptible to glucose deprivation or glycolytic inhibitors[22,23]. To investigate whether the augmented tumorigenicity of KMT2D-deficient HNSCC cells was reliant on the dysregulated activation of glycolysis, we subjected both KMT2D-KO and KMT2D-WT HNSCC cells to glucose deprivation. Glucose deprivation induced robust cell death in both KMT2D-KO SCC23 and SCC1 cells compared to their parental KMT2D-WT cells (Supplementary Fig. 3a, b), indicating the potential dependence of KMT2D-deficient HNSCC on glycolysis. Flow cytometry analysis showed that glucose deprivation induced more pronounced apoptosis in KMT2D-KO SCC23 cells (Supplementary Fig. 3c). The induction of activated Caspase-3 and cleaved PARP further confirmed the increased apoptosis in KMT2D-KO cells under glucose deprivation (Supplementary Fig. 3d, e).

Next, we treated cells with 2-deoxy-D-glucose (2-DG), the most widely used glycolytic inhibitor in both experimental and clinical oncology. As expected, 2-DG treatment led to a significant decrease in the cell viability of KMT2D-KO SCC23 cells compared with KMT2D-WT SCC23 cells (Supplementary Fig. 3f). Moreover, 2-DG treatment inhibited the subcutaneous tumor growth of KMT2D-KO SCC23 cells in nude mice (Supplementary Fig. 3g, h). Similar results were observed in KMT2D-KO SCC1 cells (Supplementary Fig. 3i, j). However, we did not observe any notable differences between the 2-DG treatment and the vehicle group in nude mice bearing KMT2D-WT SCC23 cells (Supplementary Fig. 3k, l). We also examined the inhibitory effect of 2-DG on SCC46 and SCC74A, two HNSCC cell lines harboring KMT2D heterozygous truncating mutations, and SCC9 harboring KMT2D wildtype[24]. 2-DG treatment significantly decreased the cell viabilities of SCC46 and SCC74A cells without affecting SCC9 cells in vitro (Supplementary Fig. 3m). In addition, using the tongue orthotopic xenograft mouse model, we found that 2-DG reduced the tumor volume of SCC46 cells but had no inhibitory effect on the tumorigenicity of SCC9 cells in vivo (Supplementary Fig. 3n). Importantly, consistent with the increased metastasis in *Kmt2d*-HT mice, transwell invasion assays showed that KMT2D loss significantly increased SCC23 invasiveness, and 2-DG treatment was able to reverse the enhanced invasion in KMT2D-KO SCC23 cells (Supplementary Fig. 3o). These findings indicate that

enhanced glycolysis contributes to the aggressive tumorigenicity of KMT2D-deficient HNSCC, and the inhibition of glycolysis preferentially impedes the growth of KMT2D-deficient HNSCC cells.

## KMT2D deficiency causes cell cycle arrest and FA pathway impairment upon glycolytic inhibition

While 2-DG treatment could induce apoptosis in KMT2D-KO SCC23 cells, we noticed that the surviving KMT2D-KO SCC23 cells could rapidly recover after 2-DG was removed, suggesting that KMT2D-mutant HNSCC might develop resistance to 2-DG (Supplementary Fig. 3p). In fact, based on Warburg's findings, clinical trials have been performed to determine whether 2-DG as a monotherapy treatment inhibited tumor growth in the late 50 s. However, the inhibitory effects on tumor growth were disappointing. Therefore, to avoid the adaptation and resistance of cancer cells to glycolytic inhibitors, it has been proposed to combine 2-DG with already-approved chemotherapeutic drugs for cancer treatment[25-27]. Because KMT2D-deficient HNSCC led to abnormal gene expression profiles to enhance glycolysis, we hypothesized that the glycolytic suppression may cause abnormal molecular and epigenetic changes in KMT2D-mutant HNSCC. Identifying those changes could reveal potential therapeutic vulnerabilities which might help to develop targeted therapy for KMT2D-mutant HNSCC. Therefore, we performed RNA-seq of glucose-deprived KMT2D-KO and KMT2D-WT SCC23 cells. Interestingly, Kyoto Encyclopedia of Genes and Genomes (KEGG) analysis showed that the FA pathway was significantly downregulated in KMT2D-KO cells compared with KMT2D-WT cells upon glucose deprivation (Fig. 3a). The FA pathway is critical for DNA repair and the impairment of this pathway leads to DNA repair defects[28]. Consistent with the KEGG analysis result, GO analysis revealed that the downregulated genes by the loss of KMT2D were enriched in DNA repair and replication (Fig. 3b). In addition, RNA-seq transcriptomic analysis revealed that a set of the FA genes were downregulated due to KMT2D loss upon glucose deprivation (Fig. 3c). Consistently, GSEA analysis also showed that the FA genes were downregulated due to KMT2D loss upon glucose deprivation (Fig. 3d). We confirmed the downregulation of multiple FA genes, including *ATR*, *BRCA1*, *FAN1*, *FANCD2*, *FANCI*, *FANCM*, *REV3L*, *POLQ*, and *TOP3A*, in glucose-deprived KMT2D-KO SCC23 cells by qRT-PCR (Fig. 3e). In line with the mRNA levels, the protein levels of FANCD2 and FANCI were decreased in KMT2D-KO SCC23 cells upon glucose deprivation (Fig. 3f). Instead, the protein levels of the non-affected FA genes by KMT2D loss, such as FANCG and FANCL, were not changed upon glucose deprivation (Supplementary Fig. 4a). Intriguingly, we did not detect significant changes in these FA genes due to KMT2D loss under glucose-sufficient conditions (Supplementary Fig. 4b). Consistent with the results obtained upon glucose deprivation, we also observed the downregulation of the FA genes in 2-DG-treated KMT2D-KO SCC23 cells compared with KMT2D-WT cells (Supplementary Fig. 4c). Similarly, KMT2D loss downregulated the

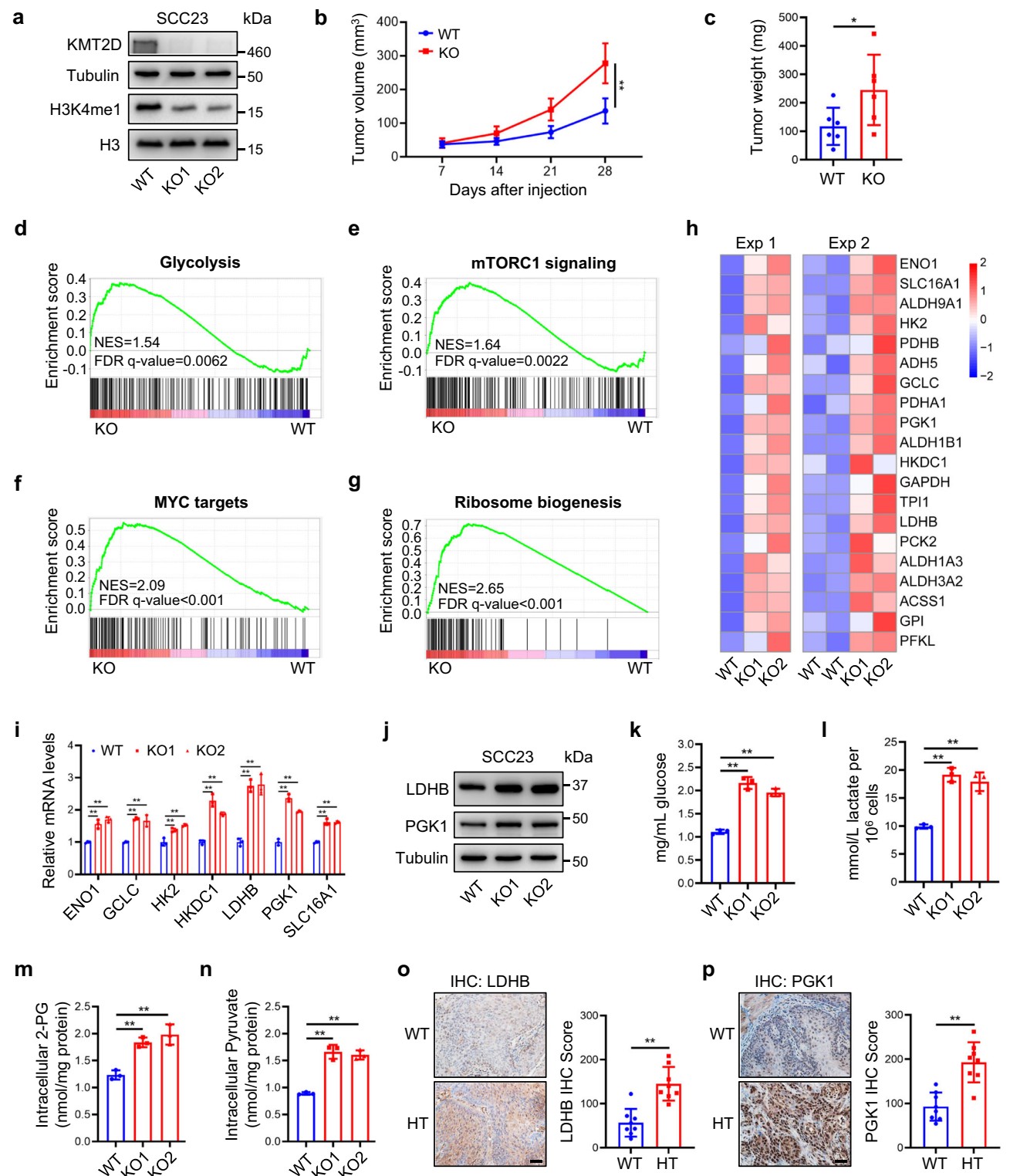

mRNA levels of multiple FA genes in glucose-deprived KMT2D-KO SCC1 cells (Supplementary Fig. 4d). FA pathway impairment increased chromosome breakage following exposure to interstrand crosslinks (ICLs) inducing agents, which is the diagnostic feature of Fanconi anemia (FA) cells[29,30]. To assess the FA pathway in KMT2D-deficient cells upon 2-DG treatment, we examined the mitomycin C (MMC)-induced chromosome breaks and radials in KMT2D-WT and KMT2D-KO SCC23 cells. The chromosomal breaks and radials in KMT2D-KO SCC23 cells were detected upon 2-DG plus MMC treatment (Fig. 3g). Qualitative measurement showed that 2-DG plus MMC treatment

significantly elevated the levels of chromosome breaks and radials in KMT2D-KO SCC23 cells compared to KMT2D-WT cells (Fig. 3h, i). The FA pathway is activated in response to DNA damage, leading to the monoubiquitination and nuclear DNA repair foci formation of FANCD2 for DNA repair[31]. Therefore, we treated KMT2D-KO and parental KMT2D-WT SCC23 cells with the DNA crosslinking inducer MMC with or without 2-DG and analyzed the formation of FANCD2 foci. In the presence of 2-DG, while MMC significantly induced the formation of FANCD foci in KMT2D-WT SCC23, MMC was unable to induce the foci formation in KMT2D-KO SCC23 cells (Fig. 3j). In contrast, in the

**Fig. 2 | Loss of KMT2D promotes HNSCC growth and upregulates glycolysis.**
**a** Protein levels of KMT2D and H3K4me1 in KMT2D-WT and KMT2D-KO SCC23 cells by western blot. $n$ = 3 independent experiments. **b, c** Xenografted tumor growth (**b**) and weight (**c**) of KMT2D-WT and KMT2D-KO SCC23 cells in nude mice. Values are mean ± SD. $n$ = 6:6. **$p < 0.01$ by two-way ANOVA with Bonferroni correction (**b**). *$p < 0.05$ by unpaired two-tailed Student's t test (**c**). **d**–**g** Enrichment plots of gene sets for glycolysis (**d**), mTORC1 signaling (**e**), MYC targets (**f**), and Ribosome biogenesis (**g**) identified by GSEA between KMT2D-KO and KMT2D-WT SCC23 cells. **h** Heatmap of glycolytic genes from KMT2D-WT and KMT2D-KO SCC23 cells by RNA-seq from 2 independent experiments. **i** mRNA levels of glycolytic genes from KMT2D-WT and KMT2D-KO SCC23 cells by qRT-PCR. Values are mean ± SD from three independent experiments. **$p < 0.01$ by one-way ANOVA. **j** Protein levels of LDHB and PGK1 in KMT2D-WT and KMT2D-KO SCC23 cells by western blot. $n$ = 3 independent experiments. **k** Glucose uptake in KMT2D-WT and KMT2D-KO SCC23 cells. Values are mean ± SD from three independent experiments. **$p < 0.01$ by one-way ANOVA. **l** Lactate excretion from KMT2D-WT and KMT2D-KO SCC23 cells. Values are mean ± SD from three independent experiments. **$p < 0.01$ by one-way ANOVA. **m,n** Intracellular levels of 2-PG (**m**) and pyruvate (**n**) in KMT2D-WT and KMT2D-KO SCC23 cells. Values are mean ± SD from three independent experiments. **$p < 0.01$ by one-way ANOVA. **o, p** IHC staining and quantification of LDHB (**o**) and PGK1 (**p**) in *Kmt2d*-WT and *Kmt2d*-HT mouse HNSCC. Scale bar, 50 μm. Values are mean ± SD. $n$ = 8:8. **$p < 0.01$ by unpaired two-tailed Student's t test. Source data are provided as a Source Data file.

absence of 2-DG, MMC was able to induce the formation of FANCD2 foci in both KMT2D-WT and KMT2D-KO SCC23 cells. In addition to DNA damage, FANCD2 can undergo monoubiquitination and nuclear foci formation during the S-phase of the cell cycle in the absence of DNA-damaging agents[32]. To explore whether 2-DG treatment affects S-phase in KMT2D-deficient SCC23 cells, we performed BrdU incorporation and propidium iodide staining. Our results showed that KMT2D loss significantly decreased BrdU incorporation in 2-DG treated KMT2D-KO SCC23 cells compared with KMT2D-WT SCC23 cells (Supplementary Fig. 4e). Moreover, KMT2D-KO SCC23 cells exhibited a significant decrease in the S-phase population compared with KMT2D-WT SCC23 cells upon 2-DG treatment by flow cytometry analysis (Supplementary Fig. 4f). Therefore, cell cycle arrest caused by KMT2D loss upon 2-DG treatment further contributed to the dysfunction of the FA pathway. These results functionally confirm our RNA-seq results, indicating that KMT2D deficiency impairs the DNA repairing function of the FA pathway in HNSCC upon 2-DG treatment.

### KMT2D-deficient HNSCC is hypersensitive to DNA damage agents under glycolytic inhibition

Based on the result that the FA pathway is impaired in KMT2D-KO HNSCC cells after treatment of 2-DG, we next examined whether the combination of 2-DG and low-dose of DNA crosslinking agents would collaboratively induce apoptosis in KMT2D-deficient HNSCC cells. Flow cytometry analysis showed that 2-DG or low-dose MMC alone induced minimal apoptosis in KMT2D-WT SCC23 cells, and their combination had weak additive effects on apoptosis. In contrast, 2-DG, but not MMC, significantly induced apoptosis in KMT2D-KO SCC23 cells, which was further significantly enhanced when MMC was added (Fig. 4a). Additionally, SCC46 cells harboring a KMT2D truncated mutation also showed enhanced sensitivity to the combination treatment (Supplementary Fig. 5a). Using the orthotopic xenograft mouse model of HNSCC, we rigidly examined the inhibitory effect of MMC plus 2-DG treatment on the tumorigenicity of SCC23 cells and SCC46 cells in vivo. In line with the in vitro results, 2-DG alone partially inhibited tumor growth of KMT2D-KO SCC23 cells and low-dose MMC did not significantly affect the tumor growth in vivo. The combination of 2-DG and low-dose MMC drastically inhibited the tumor growth of KMT2D-KO SCC23 cells (Fig. 4b). In contrast, MMC plus 2-DG treatment had insignificant effects on the tumor growth of KMT2D-WT SCC23 cells in vivo (Supplementary Fig. 5b). Consistently, MMC plus 2-DG had superior inhibitory effects on the tumor growth of KMT2D-mutant SCC46 cells compared with 2-DG or MMC treatment alone (Fig. 4c). In agreement with the tumor growth results, immunofluorescence (IF) staining of the apoptosis marker activated Caspase-3 (Ac-Casp3) showed that 2-DG alone, but not MMC, greatly induced apoptosis in SCC46 cell-derived tumors. Importantly, adding MMC with 2-DG further enhanced apoptosis in SCC46 cell-derived tumors (Fig. 4d, e). We further examined the anti-tumor ability of the combination treatment in the immunocompetent 4NQO-induced *Kmt2d*-HT mouse HNSCC model. Tamoxifen was administered at 21 weeks to delete KMT2D. At the same time, mice were treated with the vehicle,

2-DG, MMC, and 2-DG plus MMC (Fig. 4f). Consistent with the results from the orthotopic xenograft model, 2-DG combined with MMC further reduced lesion surface areas compared with 2-DG alone (Fig. 4g). Histological analysis found that MMC plus 2-DG significantly decreased HNSCC numbers, areas, and invasiveness compared with 2-DG or MMC (Fig. 4h, i). In line with the result that the FA pathway is impaired in human KMT2D-deficient HNSCC upon 2-DG treatment, IHC staining for FANCD2 in 4NQO-induced mouse HNSCC also revealed that the protein levels of FANCD2 were decreased after 2-DG treatment in *Kmt2d*-HT HNSCC (Supplementary Fig. 5c). Furthermore, anti-PCK immunostaining revealed that MMC plus 2-DG effectively eliminated lymph node metastasis compared to 2-DG or MMC alone (Fig. 4j, k).

Cisplatin, which also acts as a DNA ICL inducer, is a widely used chemotherapeutic agent for treating HNSCC. Compared with KMT2D-WT SCC23 cells, KMT2D-KO SCC23 cells were slightly resistant to cisplatin treatment in vitro under glucose-sufficient conditions (Supplementary Fig. 5d). In the presence of 2-DG, cisplatin significantly induced the formation of FANCD2 foci in KMT2D-WT SCC23, but it was unable to induce the foci formation in KMT2D-KO SCC23 cells. In contrast, in the absence of 2-DG, cisplatin was able to induce the formation of FANCD2 foci in both KMT2D-WT and KMT2D-KO SCC23 cells (Supplementary Fig. 5e). We further examined whether low-dose cisplatin plus 2-DG collaboratively inhibited KMT2D-KO SCC23 cell-derived tumor growth. We also found that low-dose cisplatin plus 2-DG potently suppressed the tumor growth of KMT2D-KO SCC23 cells compared with 2-DG or cisplatin alone (Fig. 4l). Moreover, cisplatin plus 2-DG significantly reduced the cervical lymph node metastasis of KMT2D-KO SCC23 tumors compared with 2-DG or cisplatin alone (Fig. 4m). Taken together, these in vitro and in vivo results suggest that KMT2D-deficient HNSCC is hypersensitive to the combination treatment of 2-DG and DNA crosslinking agents, MMC and cisplatin.

### KMT2D epigenetically sustains FA gene expression under glucose deprivation

To investigate how KMT2D regulates the FA genes under glycolytic inhibition, we performed chromatin immunoprecipitation sequencing (ChIP-seq) for H3K4me1, H3K27ac, H3K4me3, H3K27me3, and H3K9me3 in glucose-deprived KMT2D-KO and parental KMT2D-WT SCC23 cells to evaluate the epigenomic landscape changes caused by KMT2D loss. Using the ChromHMM algorithm, we annotated 10 different chromatin states based on combinatorial histone modification patterns and the average genome coverage by each state (Supplementary Fig. 6a). These chromatin states are as below: active promoter state including flanking transcription start site (TSS) marked by high levels of H3K4me3 (E01), active TSS marked by high levels of H3K27ac and H3K4me3 (E02), and flanking active TSS marked by high levels of H3K4me1, H3K4me3, and H3K27ac (E04); active enhancer state including active enhancer 1 with high levels of H3K27ac (E03), active enhancer 2 with high levels of H3K4me1 and H3K27ac (E05), and weak enhancer with high levels of H3K4me1 (E06); inactive chromatin state including bivalent enhancer/promoter marked by the enrichment of

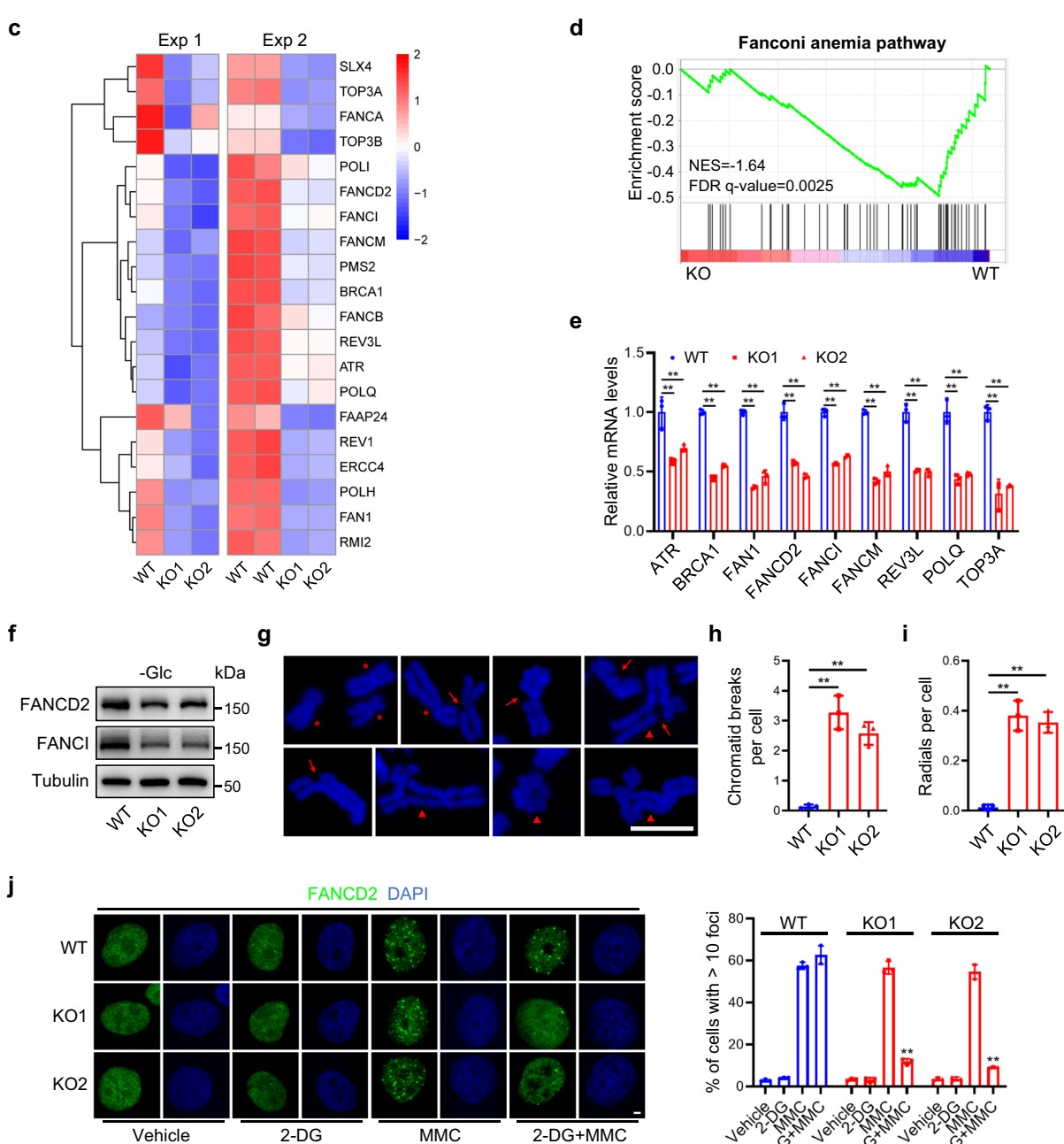

active histone modification H3K4me1 and repressive histone modification H3K27me3 (E07), repressed polycomb marked by the enrichment of H3K27me3 (E08), and heterochromatin marked by the enrichment of H3K9me3 (E09); and low state with very low signal of any detected histone modifications (E10). Next, we identified the changes in the chromatin state composition of KMT2D-KO and KMT2D-WT SCC23 cells. Considering that KMT2D works as a transcriptional co-activator and KMT2D loss directly leads to transcription

repression, we focused on the transitions from active chromatin states to inactive states caused by KMT2D loss. Interestingly, many transcriptional active regions in WT cells such as flanking TSS (E01), active TSS (E02), active enhancer 1 (E03), flanking active TSS (E04), and active enhancer 2 (E05) were transformed into inactive states, such as bivalent enhancer/promoter (E07) and heterochromatin (E09) (Supplementary Fig. 6b). These results indicate that KMT2D regulates gene transcription through both active promoters and enhancers.

**Fig. 3 | Loss of KMT2D impairs the FA pathway in HNSCC upon glycolytic inhibition. a, b** KEGG analysis (**a**) and GO analysis (**b**) of downregulated genes after glucose deprivation for 6 h in SCC23 cells from RNA-seq data. Gene expression profiles were compared between KMT2D-KO and KMT2D-WT. One-sided p-values from Fisher's Exact Test were used in the analysis. **c** Heatmap of FA genes in KMT2D-WT and KMT2D-KO SCC23 cells after glucose deprivation by RNA-seq from 2 independent experiments. **d** Enrichment plots of gene sets for the FA pathway identified by GSEA between KMT2D-KO and KMT2D-WT SCC23 cells. **e** mRNA levels of the FA pathway genes in KMT2D-WT and KMT2D-KO SCC23 cells after glucose deprivation by qRT-PCR. Values are mean ± SD from three independent experiments. **p < 0.01 by one-way ANOVA. **f** Protein levels of FANCD2 and FANCI in KMT2D-WT and KMT2D-KO SCC23 cells under glucose deprivation. *n* = 3 independent experiments. **g** Examples of chromosomal aberrations observed in 2-DG and MMC-treated KMT2D-KO SCC23 cells. Asterisks indicate normal chromatids, arrows indicate chromatid breaks, and arrowheads indicate triradial, quadriradial, and radial complex. Scale bar, 5 μm. **h, i** Quantification of chromatid breads (**h**) and radials (**i**). 50 cells were analyzed for each group. Values are mean ± SD from three independent experiments. **p < 0.01 by one-way ANOVA. **j** IF staining of FANCD2 and quantification of FANCD2 foci in SCC23 cells treated with 2-DG, MMC, or 2-DG plus MMC for 24 h. Scale bar, 2 μm. Values are mean ± SD from three independent experiments. **p < 0.01 vs WT in 2-DG plus MMC group by one-way ANOVA. Source data are provided as a Source Data file.

To further demonstrate this, we also performed ChIP-seq for KMT2D in WT cells with glucose deprivation to identify KMT2D genome-wide chromatin occupancy (Fig. 5a). The predominant chromatin states with KMT2D binding peaks in HNSCC cells upon glucose deprivation were active enhancer 1 (E03, 34.72%), active enhancer 2 (E05, 33.23%), and weak enhancer (E06, 8.18%) (Fig. 5b). Intriguingly, we found that active promoter states such as active TSS (E02) and flanking active TSS (E04) also had KMT2D binding peaks with a percentage of 11.79% and 2.08%, respectively (Fig. 5b). Moreover, ChIP-seq analysis showed that KMT2D loss significantly decreased the levels of H3K4me1 and H3K4me3 at KMT2D peak regions in HNSCC cells under glucose deprivation (Fig. 5c, d). KMT2D loss also significantly reduced H3K27ac intensity at the KMT2D peak regions (Fig. 5e), consistent with the previous findings that KMT2D is required for CBP/P300-mediated H3K27ac[9,10].

To reveal the genes directly regulated by KMT2D under glycolytic inhibition, we performed an intersection of KMT2D-bound genes identified by ChIP-seq with downregulated genes in KMT2D-KO SCC23 cells according to RNA-seq data (Fig. 6a). A total of 939 direct target genes regulated by KMT2D were identified and GO analysis of these genes revealed a direct role for KMT2D in regulating the FA pathway (Fig. 6a, b). Next, we focused on the effect of KMT2D loss on the epigenetic alterations of the FA pathway genes. Strong peaks of KMT2D, H3K4me1, H3K27ac, and H3K4me3 were found around the TSS region of the FA genes, such as *ATR*, *FANCM*, *REV3L*, and *TOP3A* (Fig. 6c–f). In contrast, no apparent occupancies of H3K9me3 or H3K27me3 were observed around the TSS region of these genes. Moreover, strong peaks of KMT2D, H3K4me1 and H3K27ac were also found at the distal enhancer regions of *FANCM* and *REV3L* (Supplementary Fig. 6c, d). Of note, in line with the reduction in gene expression, KMT2D loss markedly decreased the levels of H3K4me1, H3K27ac, and H3K4me3 around the TSS region of *ATR*, *FANCM*, *REV3L*, and *TOP3A* (Fig. 6c–f). The levels of H3K4me1 and H3K27ac were also significantly decreased at the enhancer regions of *FANCM* and *REV3L* in KMT2D-KO cells upon glucose deprivation (Supplementary Fig. 6c, d). Quantitative PCR of ChIP analysis (ChIP-qPCR) confirmed that KMT2D and H3K4me1 occupied the promoters of *ATR*, *FANCM*, *REV3L*, and *TOP3A*, and KMT2D loss significantly reduced the occupancies of KMT2D and H3K4me1 at the promoters of these genes in SCC23 cells under glucose deprivation (Supplementary Fig. 6e–h). Additionally, ChIP-qPCR results showed low occupancies of KMT2D and H3K4me1 at the promoters of *ATR*, *FANCM*, *REV3L*, and *TOP3A* genes in SCC23 cells under glucose-sufficient conditions (Supplementary Fig. 6e–h), which were consistent with the data that KMT2D loss had no significant effect on mRNA levels of these FA pathway genes in SCC23-KO cells without glucose deprivation (Supplementary Fig. 4b).

Next, we explored the mechanism that drives the increase of KMT2D occupancy at the FA genes upon glucose deprivation. AMPK activation has been shown to regulate solid tumor growth by overcoming metabolic stress such as glucose deprivation[33]. Indeed, we found that AMPKα was activated upon glucose deprivation or 2-DG treatment in SCC23 cells (Fig. 6g, h). To determine whether the activated AMPK promotes KMT2D occupancies at the promoter of the FA genes, a specific AMPK inhibitor Compound C (CC) was used to inhibit AMPK activity. Western blot confirmed its inhibitory effect on 2-DG-induced AMPK activation (Fig. 6i). Consistent with our hypothesis, 2-DG-induced KMT2D occupancies at the promoters of *ATR*, *FANCM*, *REV3L*, and *TOP3A* were significantly blocked by Compound C (Fig. 6j–m). As a result, the mRNA levels of *ATR*, *FANCM*, *REV3L*, and *TOP3A* were significantly decreased by 2-DG and Compound C combination treatment in KMT2D-WT SCC23 cells (Supplementary Fig. 6i). Cell viability was also significantly decreased upon the combination treatment of 2-DG and Compound C (Supplementary Fig. 6j). Taken together, these results suggest that glycolytic inhibition might activate AMPK to promote KMT2D occupancies on the FA genes. Consequently, KMT2D loss reduces KMT2D occupancy on the FA genes, thereby downregulating the expression of the FA genes in HNSCC cells upon glycolytic inhibition.

## KMT2D-deficient HNSCC is hypersensitive to PARP inhibitors plus 2-DG

KMT2D deficiency significantly downregulated FA genes such as *ATR*, *FANCM*, *BRCA1*, and *POLQ*. Interestingly, the inhibition or loss of these genes was also associated with increased sensitivity to PARP inhibitors[34–38]. PARP inhibitors were originally developed based on synthetic lethality in BRCA1/BRCA2-deficient cancer cells which have impairment for the double-strand break repair. Therefore, we next investigated whether 2-DG could serve as an adjuvant preferentially to render KMT2D-mutant HNSCC cells hypersensitive to olaparib, an FDA-approved PARP inhibitor. Cell viability analysis found that low-dose olaparib plus 2-DG significantly inhibited cell growth in KMT2D-KO SCC23 cells compared with KMT2D-WT SCC23 cells (Fig. 7a). Immunostaining revealed that, in the presence of 2-DG, olaparib was unable to induce the formation of FANCD2 foci in KMT2D-KO SCC23 cells while it significantly induced the foci formation in KMT2D-WT SCC23 cells (Supplementary Fig. 7a). While 2-DG partially inhibited tumor growth and low-dose olaparib did not affect tumor growth, olaparib plus 2-DG significantly inhibited orthotopic KMT2D-KO SCC23 tumor growth in mouse tongue (Fig. 7b). Next, we examined whether 2-DG could improve the efficacy of olaparib using the immunocompetent *Kmt2d*-HT mouse model of HNSCC (Fig. 7c). Similarly, olaparib plus 2-DG further significantly reduced lesion surface areas compared with 2-DG or olaparib alone (Fig. 7d). Histological analysis revealed that olaparib plus 2-DG also significantly decreased HNSCC numbers, areas, and invasiveness compared with 2-DG or olaparib alone (Fig. 7e, f). More importantly, olaparib plus 2-DG significantly inhibited lymph node metastasis as determined by anti-PCK immunostaining (Fig. 7g, h). Taken together, these in vitro and in vivo data suggest that KMT2D loss sensitizes KMT2D-mutant HNSCC tumors to the combination treatment of 2-DG and olaparib.

Next, we knocked down FANCD2 in KMT2D-WT SCC23 cells to further confirm that the sensitivity of KMT2D-deficient HNSCC to 2-DG plus MMC or olaparib was mainly caused by FA pathway impairment. Knockdown of FANCD2 was confirmed by the western blot

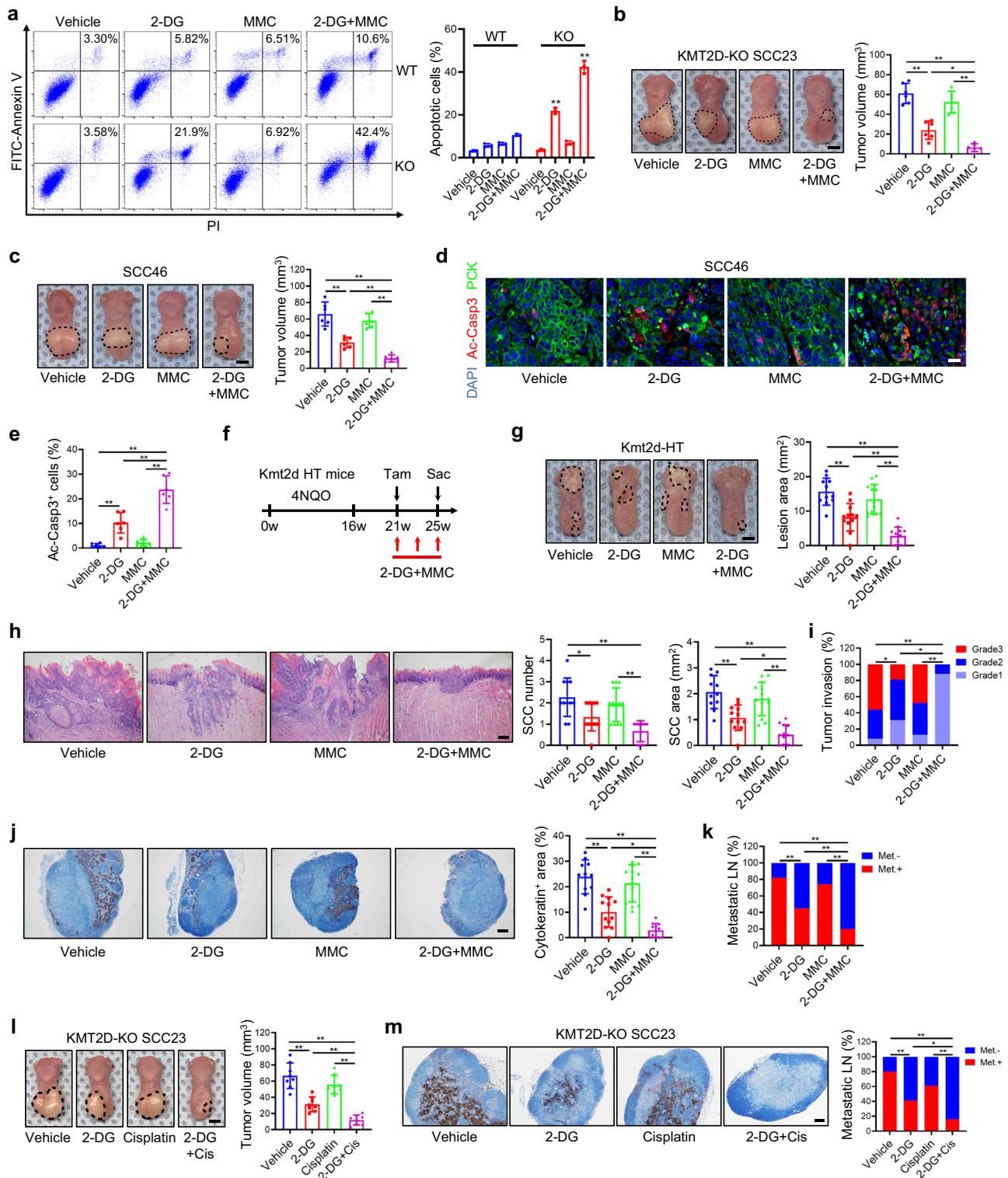

(Supplementary Fig. 7b). Flow cytometry analysis showed that there was modestly increased apoptosis in FANCD2-KD SCC23 cells compared with FANCD2-WT SCC23 cells by the treatment of 2-DG, MMC, or olaparib alone. However, the combination treatment of 2-DG plus MMC or 2-DG plus olaparib significantly induced more pronounced cell apoptosis in FANCD2-KD SCC23 cells compared with FANCD2-KD SCC23 cells (Supplementary Fig. 7c). Western blot confirmed that 2-DG plus MMC (Supplementary Fig. 7d) or 2-DG plus olaparib (Supplementary Fig. 7e) potently induced caspase-3 activation in FANCD2-KD SCC23 cells. Furthermore, while 2-DG, MMC or olaparib alone partially inhibited orthotopic tumor growth of FANCD2-KD SCC23 cells, 2-DG plus MMC or olaparib drastically inhibited the tumor growth of FANCD2-KD SCC23 cells in vivo (Supplementary Fig. 7f). We also knocked down FANCD2 in KMT2D-KO SCC23 cells (Supplementary Fig. 7g) and subjected these cells together with KMT2D-KO SCC23 cells to glucose deprivation. In contrast, FANCD2 knockdown did not further enhance glucose-deprivation-induced apoptosis in KMT2D-KO SCC23 cells (Supplementary Fig. 7h, i). As a result, 2-DG had similar inhibitory effect on orthotopic tumor growth of FANCD2-KD, KMT2D-KO SCC23 cells and KMT2D-KO SCC23 cells (Supplementary Fig. 7j).

**Fig. 4 | KMT2D-deficient HNSCC is hypersensitive to DNA cross-linking agents upon glycolytic inhibition. a** Representative scatter plots and quantification of apoptotic KMT2D-WT and KMT2D-KO SCC23 cells treated with 2-DG, MMC, or 2-DG plus MMC. Values are mean ± SD from three independent experiments. **$p < 0.01$ vs WT in the corresponding treated conditions by unpaired two-tailed Student's t test. **b,c** Volume of KMT2D-KO SCC23 (**b**) and KMT2D-mutant SCC46 (**c**) orthotopic xenografts treated with 2-DG, MMC, or 2-DG plus MMC in nude mice. Scale bar, 2 mm. Values are mean ± SD. $n = 6$ per group. *$p < 0.05$, **$p < 0.01$ by one-way ANOVA. **d,e** IF staining (**d**) and quantification (**e**) of cleaved Caspase-3 positive cells in SCC46 orthotopic xenografts. Scale bar, 20 μm. Values are mean ± SD. $n = 6$ per group. **$p < 0.01$ by one-way ANOVA. **f** Experimental design for 2-DG and MMC treatment in 4NQO-induced *Kmt2d*-HT mouse HNSCC. **g** Representative image of tongue lesions and quantification of lesion areas. Scale bar, 2 mm. Values are mean ± SD. $n = 11$:12:12:12. **$p < 0.01$ by one-way ANOVA. **h** Representative H&E staining of HNSCC and quantification of HNSCC number and area. Scale bar, 200 μm. Values are mean ± SD. $n = 11$:12:12:12. *$p < 0.05$, **$p < 0.01$ by one-way ANOVA. **i** Quantification of HNSCC invasion grades. *$p < 0.05$, **$p < 0.01$ by two-sided Cochran-Armitage test. **j** IHC staining of PCK positive cells and quantification of the metastatic area in cervical lymph nodes. Scale bar, 200 μm. Values are mean ± SD. $n = 11$:12:12:12. *$p < 0.05$, **$p < 0.01$ by one-way ANOVA. **k** Quantification of metastatic lymph nodes. **$p < 0.01$ by two-sided chi-square test. **l** Volume of KMT2D-KO SCC23 orthotopic xenografts treated with 2-DG, cisplatin, or 2-DG plus cisplatin in nude mice. Scale bar, 2 mm. Values are mean ± SD. $n = 8$ per group. **$p < 0.01$ by one-way ANOVA. **m** Representative IHC staining of PCK positive cells in cervical lymph nodes and quantification of metastatic lymph nodes. Scale bar, 200 μm. $n = 8$ per group. *$p < 0.05$, **$p < 0.01$ by two-sided chi-square test. Source data are provided as a Source Data file.

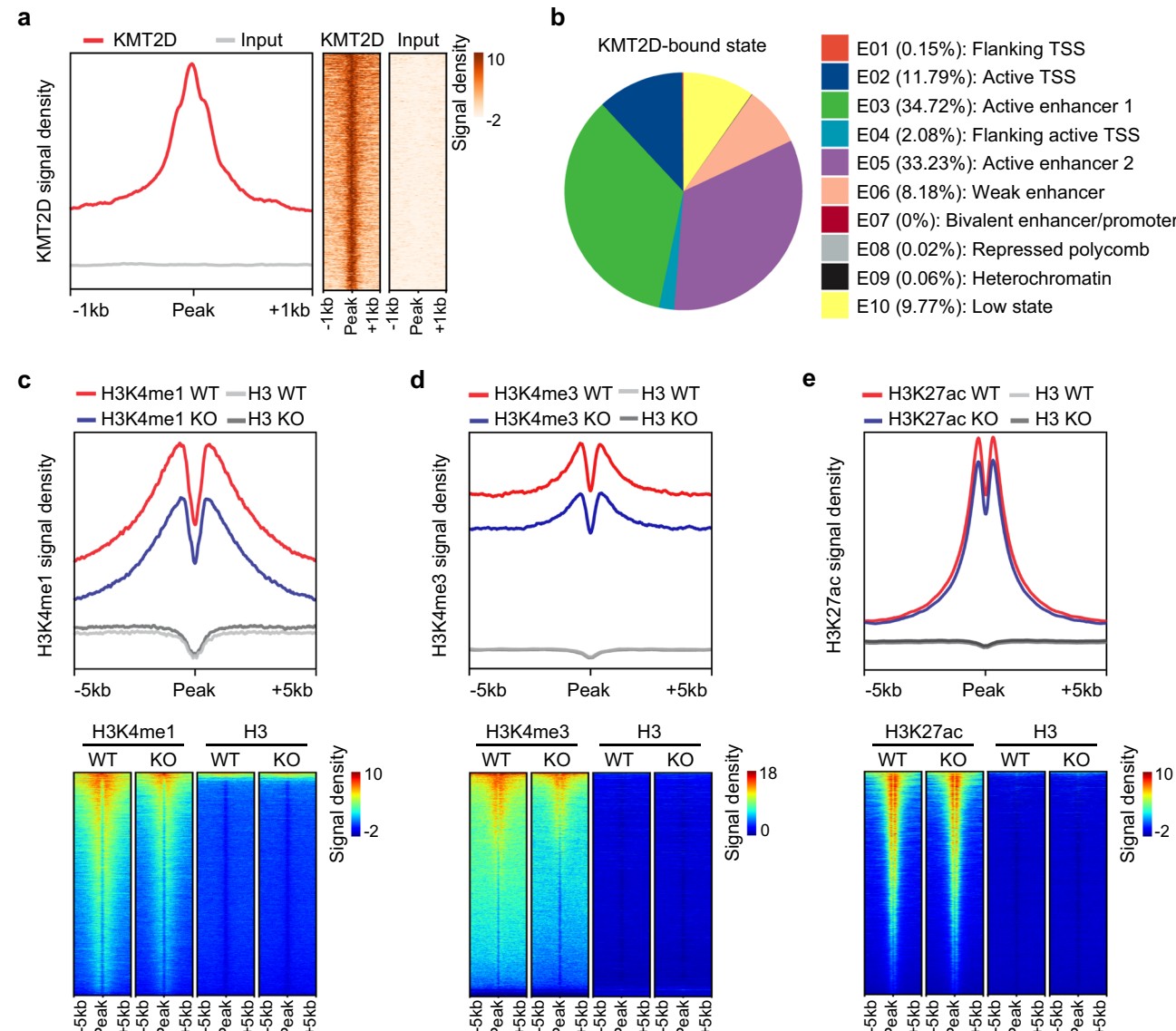

**Fig. 5 | KMT2D loss reprograms epigenomic landscape upon glycolytic inhibition. a** Average signaling intensity curves (left) and heatmap (right) of ChIP-seq reads for KMT2D in KMT2D-WT SCC23 cells after glucose deprivation. **b** Percentage of KMT2D-bound peaks at each chromatin state defined in Supplementary Fig. 6a. **c–e** Average signaling intensity curves (up) and heatmap (down) of ChIP-seq reads for H3K4me1 (**c**), H3K4me3 (**d**), and H3K27ac (**e**) at KMT2D binding region in KMT2D-WT and KMT2D-KO SCC23 cells after glucose deprivation.

To further investigate the therapeutic potential of 2-DG with DNA crosslinking agents or PAPR inhibitors, we assessed the efficacy of these combination treatments using HNSCC patient-derived xenograft (PDX) with KMT2D-inactivating mutation[39]. HNSCC PDXs with or without KMT2D mutation were subcutaneously transplanted into the flank area of NSG mice and treated as indicated. Consistently, 2-DG alone, but not low-dose cisplatin alone, partially inhibited the tumor growth of PDXs with KMT2D mutation, and cisplatin plus 2-DG nearly

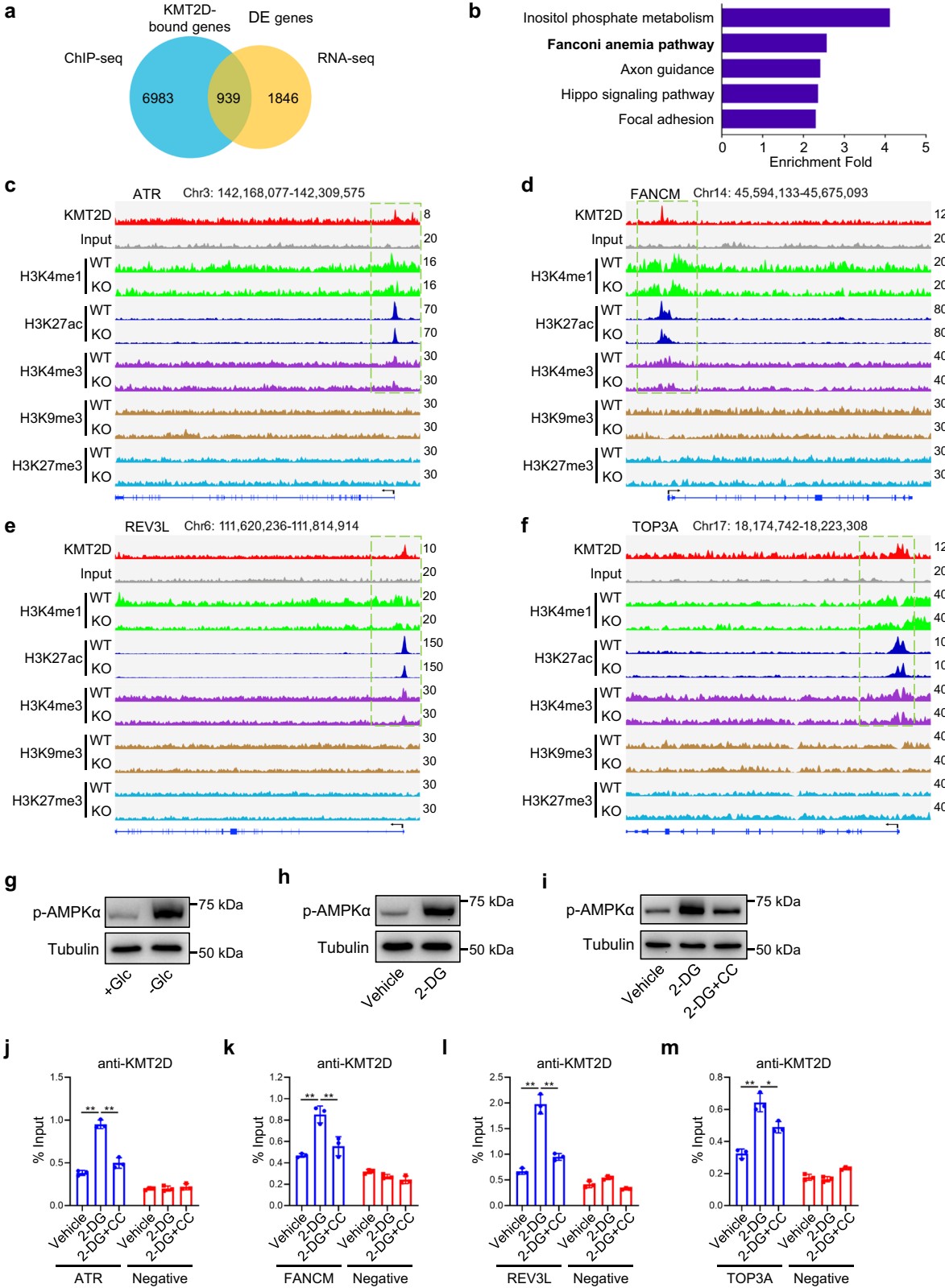

**Fig. 6 | AMPK activation increases the occupancy of KMT2D at FA genes upon glycolytic inhibition. a** Venn diagram of overlap genes between KMT2D-bound genes from ChIP-seq data and downregulated (DE) genes by KMT2D loss from RNA-seq data. **b** GO enrichment analysis of KMT2D direct targets, with top 5 biological processes were shown. **c**–**f** ChIP-seq binding signals of KMT2D in KMT2D-WT SCC23 cells and five chromatin marks (H3K4me1, H3K27ac, H3K4me3, H3K9me3, and H3K27me3) for *ATR* (**c**), *FANCM* (**d**), *REV3L* (**e**), and *TOP3A* (**f**) in KMT2D-WT and KMT2D-KO SCC23 cells under glucose deprivation. **g**–**i** Protein levels of activated AMPK under glucose deprivation (**g**), 2-DG treatment (**h**) and 2-DG or 2-DG plus Compound C treatments (**i**) in SCC23 cells by western blot. *n* = 3 independent experiments. **j**–**m** ChIP-qPCR analysis of KMT2D at *ATR* (**j**), *FANCM* (**k**), *REV3L* (**l**), and *TOP3A* (**m**) locus in SCC23 cells treated with 2-DG or 2-DG plus Compound C. Values are mean ± SD from three independent experiments. *$p < 0.05$, **$p < 0.01$ by one-way ANOVA. Source data are provided as a Source Data file.

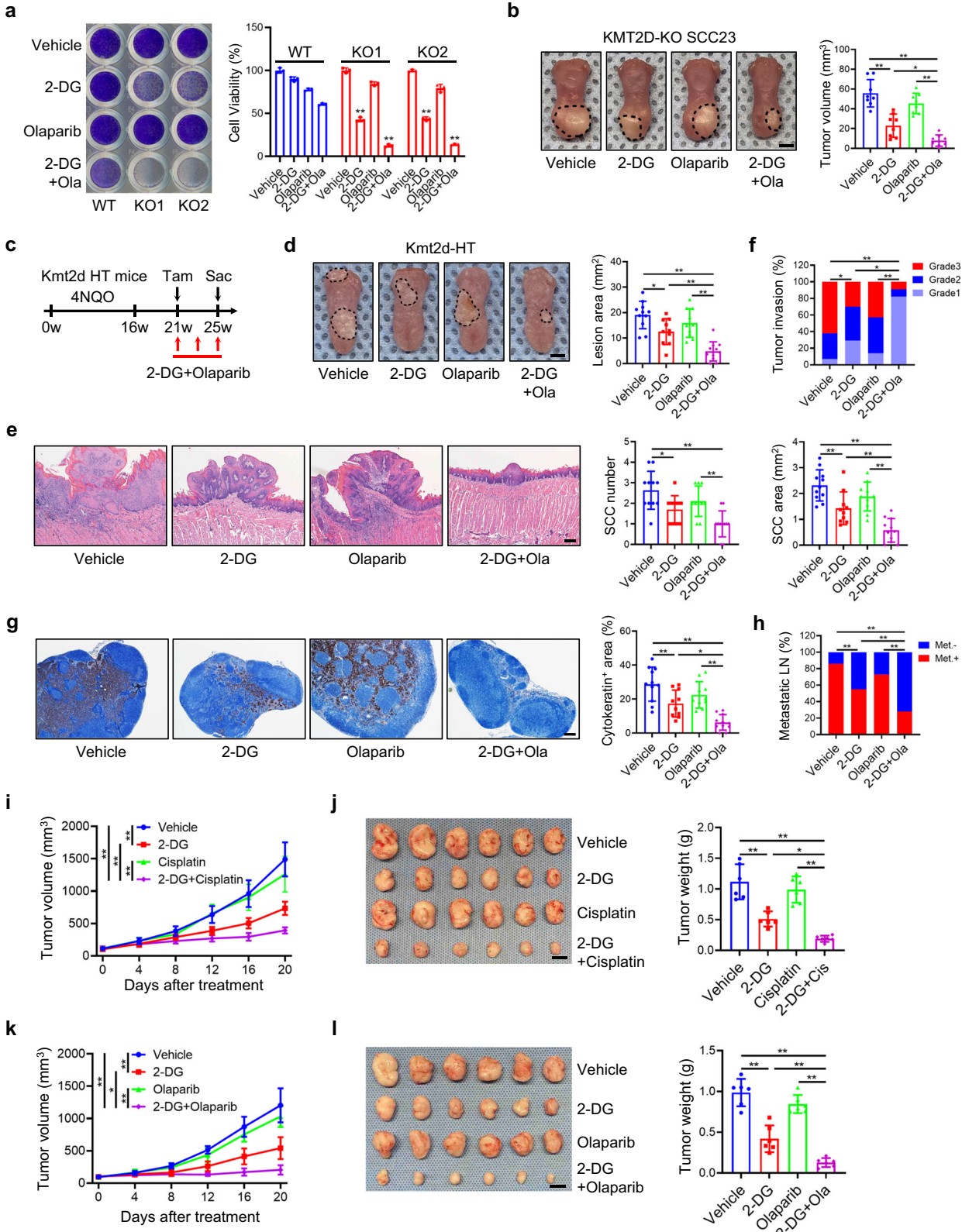

led to tumor regression in mice bearing PDXs with KMT2D mutation (Fig. 7i, j). Immunostaining showed that cisplatin plus 2-DG was unable to induce the formation of FANCD2 foci in PDXs with KMT2D mutation compared with cisplatin (Supplementary Fig. 8a). Similarly, olaparib plus 2-DG also had a superior inhibitory effect on mice bearing PDXs with KMT2D-inactivating mutation compared with 2-DG or olaparib alone (Fig. 7k, l). Olaparib plus 2-DG was unable to induce the

formation of FANCD2 foci in PDXs with KMT2D mutation compared with olaparib alone (Supplementary Fig. 8b). Similar to cisplatin, olaparib or 2-DG alone, cisplatin plus 2-DG or olaparib plus 2-DG did not significantly affect mouse body weights (Supplementary Fig. 8c, d). Furthermore, those combination treatments did not significantly inhibit tumor growth in mice bearing KMT2D-WT PDX (Supplementary Fig. 8e, f).

**Fig. 7 | KMT2D-deficient HNSCC is hypersensitive to PARP inhibitors upon glycolytic inhibition. a** Crystal violet staining and cell viability of KMT2D-WT and KMT2D-KO SCC23 cells treated with 2-DG and olaparib. Values are mean ± SD from three independent experiments. **$p < 0.01$ vs WT in the corresponding treated conditions by one-way ANOVA. **b** Volumes of KMT2D-KO SCC23 orthotopic xenografts. Scale bar, 2 mm. Values are mean ± SD. $n = 8$ per group. *$p < 0.05$, **$p < 0.01$ by one-way ANOVA. **c** Experimental design for 2-DG and olaparib treatment in 4NQO-induced *Kmt2d*-HT mouse HNSCC. **d** Representative image of tongue lesions and quantification of lesion areas. Scale bar, 2 mm. Values are mean ± SD. $n = 11{:}10{:}10{:}11$. *$p < 0.05$, **$p < 0.01$ by one-way ANOVA. **e** Representative H&E staining of HNSCC and quantification of HNSCC number and area. Scale bar, 200 μm. Values are mean ± SD. $n = 11{:}10{:}10{:}11$. *$p < 0.05$, **$p < 0.01$ by one-way ANOVA. **f** Quantification of HNSCC invasion grades. *$p < 0.05$, **$p < 0.01$ by two-sided Cochran-Armitage test. **g** IHC staining of PCK positive cells and quantification of the metastatic area in cervical lymph nodes. Scale bar, 200 μm. Values are mean ± SD. $n = 11{:}10{:}10{:}11$. *$p < 0.05$, **$p < 0.01$ by one-way ANOVA. **h** Quantification of metastatic lymph nodes. **$p < 0.01$ by two-sided chi-square test. **i, j** Tumor volumes (**i**) and weights (**j**) of KMT2D-mutant PDX treated with 2-DG, cisplatin or 2-DG plus cisplatin. Values are mean ± SD. $n = 6$ per group. Scale bar, 1 cm. **$p < 0.01$ by two-way ANOVA with Bonferroni correction (**i**). *$p < 0.05$, **$p < 0.01$ by one-way ANOVA (**j**). **k, l** Tumor volumes (**k**) and weights (**l**) of KMT2D-mutant PDX treated with 2-DG, olaparib or 2-DG plus olaparib. Values are mean ± SD. $n = 6$ per group. Scale bar, 1 cm. *$p < 0.05$, **$p < 0.01$ by two-way ANOVA with Bonferroni correction (**k**). **$p < 0.01$ by one-way ANOVA (**l**). Source data are provided as a Source Data file.

## Discussion

Our study demonstrates that KMT2D loss not only drives HNSCC initiation, but also promotes HNSCC progression and metastasis using the immunocompetent and autochthonous mouse model of HNSCC combined with spatiotemporal deletion of KMT2D. While the enhanced glycolysis played a role in the aggressive growth of KMT2D-deficient HNSCC, more importantly, our study demonstrates that KMT2D loss drastically reprogramed transcriptional and epigenomic states and led to the inhibition of the FA genes under glycolysis inhibition. The impairment of the FA pathway rendered aggressive KMT2D-deficient HNSCC hypersensitive to DNA crosslinking agents and PARP inhibitors. Based on these findings, our in vivo studies demonstrate that MMC, cisplatin or olaparib, adjuvanted with 2-DG, preferentially inhibited the growth of KMT2D-mutant HNSCC.

KMT2D is one of the most frequently mutated genes across various types of cancers. Notably, one of the frequently occurring mutations of KMT2D is truncation, which leads to catalytic inactivity and loss-of-function of KMT2D, suggesting that it may function as a tumor suppressor. Rather unexpectedly, KMT2D has divergent functions in different types of cancers and can act as a tumor suppressor or an oncogene[40–44]. KMT2D has been reported to function in a cellular context-dependent manner by interacting with specific transcription factors[11]. In this study, we showed that KMT2D loss significantly upregulates glycolytic gene expression and promotes HNSCC tumorigenesis through increasing Warburg effect. Notably, other studies also report that KMT2D loss increases glycolysis in lung cancer, melanoma, and pancreatic cancer[15–17]. Our work here, together with others, indicates that KMT2D has a fundamental role in repressing the Warburg effect. As a transcriptional co-activator, KMT2D loss will lead to the downregulation of its direct target genes, which means that KMT2D indirectly regulates glycolytic genes. It has been reported that KMT2D loss directly decreases the expression of *PER2* or *IGFBP5* in lung cancer and melanoma respectively[15,16], which leads to the subsequent increased glycolysis. Although we did not find that KMT2D loss in HNSCC affected the expression of PER2 or IGFBP5, it is most likely that KMT2D mutation utilizes a similar mechanism to promote glycolytic gene expression. In the future, it is important to identify potential transcription repressors of glycolytic genes in HNSCC which are regulated by KMT2D. In addition to enhanced glycolysis, we found that KMT2D loss also promoted mTOC1 signaling, MYC activation and ribosome synthesis. Additionally, it has been reported that KMT2D plays an important role in modulating the immune microenvironment and immune checkpoint blockade[45,46]. In the future, it will be interesting to examine the role of these pathways in KMT2D mutation-induced HNSCC and explore the potential effect of KMT2D mutation on immunotherapy using spatial transcriptomics and single-cell RNA sequencing. Unexpectedly, we found that, under glycolysis inhibition, KMT2D loss causes a significant downregulation of multiple genes in the FA pathway. Further studies suggest that glycolytic inhibition activated AMPK to promote KMT2D occupancies at the promoters of the FA genes in HNSCC cells. However, how AMPK promotes KMT2D

binding to the promoters of the FA genes is unknown. It has been reported that the phosphorylation of KMT2D can affect its methyltransferase activity and occupancy at specific target loci[47]. In the future, it will be interesting to examine whether AMPK directly phosphorylates KMT2D and then regulates its occupancies in the FA genes. Nevertheless, our results suggest that, in addition to regulating gene expression in a tissue-specific manner, KMT2D may also regulate gene expression in a metabolic state-specific manner, which highlights the divergent regulatory functions of KMT2D.

As a histone lysine methyltransferase, KMT2D directly catalyzes the mono-, di-, and tri-methylation at H3K4. In addition, KMT2D is also required for the formation of H3K27ac mediated by CBP/p300[10]. The predominant regulatory way of KMT2D that has been reported is through activating enhancers/super-enhancers, which is marked by the presence of H3K4me1 and H3K27ac. For example, KMT2D activates a subset of KMT2D-bound enhancers and upregulates genes critical for cell migration in *NRAS*-mutated melanomas[48]. In addition to enhancers/super-enhancers, other studies report that KMT2D also activates gene expression by upregulating sharp H3K4me3 peaks at gene promoters or broad H3K4me3 peaks[49–51]. In this study, we analyzed the binding region of KMT2D in glucose-deprived HNSCC cells and the epigenomic landscape changes caused by KMT2D loss. Consistent with previous studies, we found that the majority of binding regions of KMT2D in glucose-deprived HNSCC cells are active enhancers. Interestingly, binding peaks of KMT2D on active promoter regions near the TSS are also observed, even though to a lesser extent. Moreover, active enhancers and promoters were transformed into inactive states such as bivalent enhancers/promoters or heterochromatin by KMT2D loss. Therefore, our findings indicate that KMT2D regulates gene transcription through both active promoters and enhancers in HNSCC. Interestingly, inconsistent with previous studies showing that H3K4me1 peaks usually occupy enhancer regions, our ChIP-seq data showed that KMT2D regulated the FA gene expressions by modulating H3K4me1, H3K27ac, and H3K4me3 levels at the promoters near TSS in HNSCC cells under glycolytic suppression, revealing another regulatory mechanism by which KMT2D controls gene expression.

The FA pathway is an important DNA repair pathway that recognizes DNA damage and orchestrates DNA damage responses, especially for DNA ICL repair[52,53]. Cells with FA pathway deficiency are hypersensitive to DNA crosslinking agents, such as MMC and cisplatin[54]. Our study shows that, upon glucose deprivation, KMT2D loss leads to a significant downregulation of multiple genes belonging to the FA pathway. As expected, combination treatment of 2-DG and MMC or cisplatin preferentially suppresses the tumor growth and metastasis of KMT2D-mutant HNSCC. Moreover, a growing body of evidence shows that mutations of several FA pathway genes were associated with increased sensitivity to PARP inhibitors. For example, BRCA1 or BRCA2 dysfunction remarkably sensitizes cells to PARP inhibitors[36]. POLQ inhibitors have been reported to elicit synthetic lethality with PARP inhibitors[37,38]. Moreover, FANCM deficiency or ATR inhibition also rendered tumor cells hypersensitive to PARP

inhibitors[34,35]. In line with the results that KMT2D loss significantly reduced the expression levels of those genes in glucose-deprived HNSCC, the combination treatment of 2-DG and olaparib efficiently impeded the tumor growth of KMT2D-deficient HNSCC cells. While mutations of some FA genes were associated with increased sensitivity to PARP inhibitors, it was also shown that defects in upstream FA genes in HNSCC cells did not confer hypersensitivity to PARP inhibitors[35]. In addition to the FA pathways, ATR, BRCA1, and POLQ play a critical role in DNA damage repair independently. Our findings could not rule out that the hypersensitivity of KMT2D-mutant HNSCC cells to PARP inhibitors was due to the defects in other DNA damage repair mechanisms by the inhibition of ATR, BRCA1 and POLQ. Alternatively, it is also possible that the hypersensitivity to PARP inhibitors could be caused by the combined effect of the defects in both FA and other pathways.

In summary, our study demonstrates that KMT2D functions as a tumor suppressor in HNSCC. More importantly, our study demonstrates that combining 2-DG with MMC, cisplatin or olaparib leads to enhanced anti-tumor efficiency preferentially in HNSCC harboring KMT2D-inactivating mutations. As the most widely used glycolytic inhibitor, the safety and efficacy of 2-DG mono-therapy or combination with other anti-tumor therapies have been demonstrated by clinical trials[55]. Cisplatin-based treatment is the standard treatment for HNSCC and olaparib has also been successfully and clinically utilized in treating BRCA1/2-deficient ovarian and breast cancers[56–58]. Our findings mechanistically reveal that combining 2-DG with cisplatin or PARP inhibitors will benefit HNSCC patients with KMT2D-inactivating mutations, providing an important epigenomic basis for designing clinical trials for this patient cohort.

## Methods
### Mice
*Kmt2d[f/f]* (JAX:032152) mice, *K14[CreER]* (JAX:005107) mice, nude mice (JAX:002019), and NSG mice (JAX:005557) were purchased from the Jackson Laboratory. *K14[CreER]* mice were backcrossed to C57BL/6 J for five generations and then were crossed with *Kmt2d[f/f]* mice to generate *K14[CreER];Kmt2d[f/+]* mice. For in vivo heterozygous deletion of *Kmt2d*, *K14[CreER];Kmt2d[f/+]* mice were intraperitoneally administered three consecutive injections of 70 mg/kg body weight tamoxifen at the indicated time. Male and female littermate mice were used and divided into different experimental groups for the 4NOQ HNSCC model. Female nude mice were used for the orthotopic HNSCC injection, and female NSG mice were used for human HNSCC PDX xenografts. All mice were fed on a chow diet (LabDiet: 5053 PicoLab Rodent Diet 20). The maximum tumor size did not exceed 1.5 cm as determined by measuring the largest diameter, and mice were euthanized with $CO_2$ inhalation before the tumor exceeded this allowable size. All mice were maintained at UCLA in a specific pathogen-free (SPF) animal facility under 12-h light/12-h dark cycle at room temperatures ranging between 20 °C and 26 °C and humidities between 30% and 70%. All the experiments were carried out using the protocol approved by the UCLA Animal Research Committee (Protocol Number#ARC-2007-062).

### Cell lines
Human SCC23, SCC1, and SCC9 KMT2D-wildtype HNSCC cell lines were obtained from the University of Michigan at Ann Arbor, and KMT2D-mutant SCC46 and SCC74A cells were from Drs. Zhong Chen and Carter Van Waes at the National Institute on Deafness and Other Communication Disorders, National Institute of Health, with permission from Dr. Thomas E. Carey at the University of Michigan at Ann Arbor. These cell lines were maintained in high glucose DMEM containing 10% FBS and 1% antibiotics at 37 °C in a 5% $CO_2$ atmosphere. SCC cell lines with KMT2D mutations were confirmed by PCR-based sequencing, and the SCC1 and SCC23 cell lines that carry the majority of frequent and pathogenic single nucleotide variants in HNSCC were

determined by whole exome sequencing[59]. Cell lines were routinely tested negative for mycoplasma contamination using the Universal Mycoplasma Detection Kit (ATCC, Cat#30-1012 K).

### 4NQO mouse HNSCC model
For induction of HNSCC, six-week-old mice were treated with 40 μg/mL 4NQO-containing drinking water for 16 weeks and then normal drinking water for another 9-10 weeks for tumor formation and lymph node metastasis. For 2-DG, MMC, olaparib or their combination treatment, *Kmt2d*-HT mice bearing HNSCC were randomly separated into four groups and given: 1) control vehicle; 2) 2-DG (500 mg/kg); 3) MMC (1 mg/kg) or olaparib (60 mg/kg); 4) 2-DG plus MMC or olaparib as indicated. The treatment was given every two days. Mice were sacrificed, and tongues and cervical lymph nodes were collected immediately. The surface lesion areas of tongue tumors were measured. Longitudinally cut tongues (dorsal/ventral) and intact lymph nodes were fixed in 10% buffered formalin for 48 h, and then paraffin-embedded, section cut processed, and H&E stained by the UCLA Translational Pathological Core. The SCC numbers, areas, and invasion depth were determined according to H&E staining using Cellsens software. The percentage of lymph nodes with metastasis and metastatic areas was measured based on the IHC staining of PCK.

### RNA isolation, qRT-PCR and RNA-seq
Total RNA was extracted using TRIzol reagents. cDNA was generated using the reverse transcription reaction with 1 μg RNA, Random Hexamers, dNTP mix, and M-MuLV Reverse Transcriptase. Real-time PCR was performed using SYBR Green supermix on a Bio-Rad CFX96 machine. Relative expression levels of indicated genes were compared with B2M expression using $2^{-\Delta\Delta Ct}$ method. The primers used for qRT-PCR are listed in Supplementary Table 1. For RNA-seq, RNA quality control, library construction, and sequencing were all performed by the Technology Center for Genomics & Bioinformatics (TCGB) core at UCLA, and libraries were sequenced on Illumina HiSeq 3000. Analysis of RNA-seq data was done as described before[59]. Briefly, RNA-seq FASTQ sequences were mapped to the human genome hg38 (GRCh38) by Hisat2 (v2.2.0). For the uniquely mapped reads, FeatureCounts from Subread2 (v2.0.3) were used to count the reads mapped to each gene. The differentially expressed genes (DEGs) were identified by Deseq2 (v1.32.0). Gene ontology (GO) term and gene set enrichment analysis (GSEA) were utilized to find enriched functional annotations for differentially expressed genes. GO term analyzes were carried out for gene transcripts both upregulated and downregulated by at least 1.5-fold. GO analysis was carried out using a database for annotation, visualization, and integrated Discovery (DAVID) version 6.8. GSEA was performed by using the annotated gene sets in the molecular signatures database version 7.1.

### ChIP-seq and data analysis
ChIP procedures were performed using Pierce™ Magnetic ChIP Kit (Cat#26157) following the manufacturer's instructions. The antibodies used for ChIP were: Anti-KMT2D (Sigma-Aldrich, ABE1867), H3K4me1 (Abcam, ab8895), H3K27ac (Active Motif, 39133), H3K4me3 (Abcam, 8580), H3K9me3 (Abcam, 8898), H3K27me3 (Abcam, 6002), H3(Abcam, ab1791). All the antibodies were diluted 1:100 and incubated with chromatin overnight. The purified DNAs were proceeded to libraries construction and sequencing performed by TCGB core at UCLA using Illumina HiSeq 3000 or Novaseq SP. All ChIP-Seq sequencing reads were mapped to the human genome version hg19 (GRCh37) by Bowtie2 (v2.4.2). For the KMT2D binding and histone modification (H3K4me1, H3K4me3, H3K27ac, H3K9me3, and H3K27me3) peaks calling, only uniquely mapped reads were used for peak calling with MACS2 (v2.2.8) with p value less than 0.01. Deeptools (v2.0) were used for visualization of the histone modification intensity at the KMT2D peaks.

## Chromatin annotation and enrichment analysis

Chromatin State Modeling was performed with ChromHMM (v1.24) by integrating the histone modification peaks (H3K4me1, H3K4me3, H3K27ac, H3K9me3, and H3K27me3) in both KMT2D-WT and KMT2D-KO group, respectively. To choose the state number, we first modeled all the chromatin states from 5 states to 15 states. By comparing the chromatin annotation of predicted states in different models, 10 states HMM models were selected, and we annotated the chromatin states based on the histone modification and genome distribution. To identify the ChromHMM transition between KMT2D-WT and KMT2D-KO group, the chromatin states identified in the KO group were mapped to WT ChromHMM annotations using the BEDtools intersect function. Next, OverlapEnrichment was used to perform an enrichment test for the chromatin state of the KO mapped on the chromatin states of the WT. The matrix output from OverlapEnrichment was scaled by columns and plotted using Pheatmap.

## Immunohistochemistry and immunofluorescence staining

For all collected specimens, 5 μm paraffin-embedded sections were sequentially deparaffinized and rehydrated. For IHC staining, after antigens retrieval (Agilent Dako, Cat#S1699) and endogenous peroxidase and alkaline phosphatase blocking (LSBio, Cat#LS-J1031), sections were incubated with primary antibodies overnight at 4 °C. After washing, the sections were incubated with secondary antibodies (Agilent Dako, Cat#K4001/K4003) and the signals were detected with liquid DAB+ substrate chromogen system (Agilent Dako, Cat#K3468), then counterstained with hematoxylin and mounted with aqueous permanent mounting medium (Agilent Dako, Cat#S196430-2). For IF staining, the sections were incubated with fluorochrome-conjugated secondary antibodies after primary antibody incubation, then mounted with ProLong Diamond Antifade Mountant with DAPI. Representative images were acquired with microscopy using Cellsens software. The percentages of Ki-67, Cleaved Caspase-3, PCK, and FANCD2 puncta-positive cells from 5 fields were counted and averaged.

## BrdU labeling and detection

5-Bromo-2′-deoxyuridine (BrdU) immunofluorescence was performed to assess cell proliferation upon 2-DG treatment. In brief, after 24 h of treatment with 2-DG, KMT2D-WT and KMT2D-KO SCC23 cells were incubated with 25 μM BrdU (Thermo Scientific, Cat#000103) for 4 h. The cells were fixed with 4% paraformaldehyde for 20 min at 4 °C and later treated with 1 N HCl for 15 min at 37 °C and neutralized with 0.1 M borate buffer (pH 8.5) for 10 min at room temperature. Subsequently, the cells were subjected to an immunofluorescence staining procedure with anti-BrdU specific primary antibodies (Abcam, Cat#ab6326).

## Chromosome breakage and radial formation

To quantify breakages and radial formations, cells were seeded in 6-well plates at $2 \times 10^5$ cells/well. After attached, cells were treated with 1 mM 2-DG and 20 ng/mL MMC for 24 h and then treated with 0.2 μg/mL colcemid (Thermo Scientific, Cat#15212012) for another 2 h. Cells were harvested and resuspended in 75 mM KCl for 15 min at 37 °C. Next, cells were fixed with methanol: acetic acid (3:1). The fixing step was repeated twice. Cells were then dropped onto glass slides and air-dried overnight before coverslips were mounted onto the slides with ProLong Diamond Antifade Mountant containing DAPI. Images of metaphase spreads were acquired and assessed using a Leica SP5X laser scanning confocal microscope. Visible chromosome breaks and radials were scored[29], and analysis was performed using GraphPad Prism (v9.4.0).

## HNSCC PDX model, subcutaneous and orthotopic HNSCC tumor growth

KMT2D wildtype PDX (ID:TM01141) was purchased from the Jackson Laboratory, and KMT2D nonsense mutation PDX was provided by Dr. Antonio Jimeno from the University of Colorado School of Medicine[39]. For the PDX xenograft study, the PDX tumors were chopped into small fragments and subcutaneously transplanted into the flank area of 6-week-old female NSG mice using a 13 g cancer implant needle. For SCC23 and SCC1 subcutaneous xenograft models, $2 \times 10^6$ cells were mixed with Matrigel at a 1:1 ratio and injected subcutaneously into the flank area of 6-week-old female nude mice. For SCC23 and SCC46 orthotopic xenograft models, $5 \times 10^5$ cells were mixed with Matrigel at a 1:1 ratio and submucosally injected into the tongue of 6-week-old female nude mice. For combination treatment, tumor-bearing mice were randomly separated into four groups and given the following chemicals every two days: 1) control vehicle; 2) 2-DG (500 mg/kg); 3) MMC (1 mg/kg), cisplatin (1 mg/kg) or olaparib (60 mg/kg); 4) 2-DG plus MMC, cisplatin or olaparib. Tumor volume was calculated using the volume formula for an ellipsoid: $1/2 \times D \times d^2$ where $D$ is the longer diameter and $d$ is the shorter diameter.

## Western blot

Total proteins were extracted using RIPA lysis buffer supplemented with a cocktail of protease inhibitors and phosphatase inhibitors. 50 μg of proteins were electrophoresed by SDS-PAGE and then transferred to a PVDF membrane using the semi-dry transfer apparatus. After blocking with 5% milk, membranes were incubated with primary antibodies at 4 °C overnight. Membranes were then washed and incubated with HRP-conjugated secondary antibodies for 1 h at RT. Signals were revealed by Clarity Western ECL Substrate. The primary antibodies used for western blot were: Anti-KMT2D (Sigma-Aldrich, ABE1867, 1:500), LDHB (Santa Cruz Biotechnology, sc-100775, 1:1000), PGK1 (Santa Cruz Biotechnology, sc-130335, 1:1000), H3K4me1 (Abcam, ab8895, 1:10000), α-tubulin (Sigma-Aldrich, T5168, 1:10000), Cleaved Caspase-3 (Cell Signaling Technology, 9661, 1:1000), Cleaved PARP (Cell Signaling Technology, 5625, 1:1000), FANCD2 (Abcam, ab108928, 1:2000), FANCI (Abcam, ab245219, 1:5000), FANCG (Thermo Fisher Scientific, 10215-1-AP, 1:1000), FANCL (Proteintech, 66639-1-Ig, 1:1000), KMT2A (Novus Biologicals, NB600-248, 1:1000), KMT2B (Cell Signaling Technology, 47097, 1:1000), KMT2C (Sigma-Aldrich, SAB1300082, 1:1000), Phospho-AMPKα (Cell Signaling Technology, 2535, 1:1000). Uncropped versions of the blots were provided in the Source Data file.

## CRISPR-Cas9 mediated gene editing and gene knockdown by shRNA

To generate KMT2D knockout cell lines, SCC23 and SCC1 cells were transfected with pSpCas9(BB)-2A-GFP plasmid (Addgene, Cat#48138) containing sgRNA targeting either LacZ or KMT2D for 24 h. GFP-positive cells were then sorted by flow cytometry and sorted cells were cultured in the normal growth medium. The gating strategy is provided in Supplementary Fig. 9. To verify the successful knockout of KMT2D, genomic DNA from these cells was extracted and analyzed by PCR and Sanger sequencing to detect CRISPR-induced mutations. In addition, KMT2D knockout in these cells was further confirmed by Western blot.

To generate lentiviruses expressing FANCD shRNA, lentiviral shRNA control plasmid (sh-Ctrl, Horizon Discovery, RHS6848) and *FANCD2* specific shRNA lentiviral plasmid (sh-FANCD2, Horizon Discovery, RHS3979-201909896) were transfected into 293 T/17 cells (ATCC, Cat#CRL-11268) with two helper plasmids psPAX2 (Addgene, Cat#12260) and pMD2.G (Addgene, Cat#12259). Viral supernatants were harvested 48 h and 72 h after transfection, filtered, and immediately used for infection with 8 μg/mL polybrene (Sigma-Aldrich, Cat#TR-1003-G). 48 h after infection, cells were selected with puromycin (Invivogen, Cat#ant-pr-1) at 2 μg/mL for 5 days. The knockdown of FANCD2 was confirmed by Western blot and qRT-PCR.

## Glucose uptake, lactate production, and intracellular glycolytic metabolites detection

A total of $1 \times 10^6$ KMT2D-WT and KMT2D-KO SCC23 cells were seeded into each well of 6-well plates in a 2 ml complete culture medium. Cells were incubated for 24 h, and the culture medium was then collected to measure the glucose and lactate concentrations. Glucose levels were determined using a glucose (GO) assay kit (Sigma, Cat#GAGO20-1KT). Glucose uptake was defined as the difference in glucose concentration in the medium with or without cell incubation. Lactate levels were determined using the Lactate Colorimetric/Fluorometric Assay Kit (BioVision, Cat#K607-100). Cells were collected and counted, and lactate production was normalized according to the cell numbers. Intracellular concentrations of 2-phosphoglycerate (2-PG) and pyruvate in HNSCC cells were detected using the glycolytic metabolite detection kits (Abcam, Cat#ab174097 and Cat#ab65342, respectively) according to the manufacturer's instructions.

## Cell apoptosis and cell viability assay

HNSCC cells were treated with 2-DG (1 mM), MMC (20 ng/ml), cisplatin (2 μM), olaparib (20 μM), and 2-DG plus MMC, cisplatin or olaparib. Chemicals are listed in Supplementary Table 2. After treatment, the apoptotic rates were measured using the FITC Annexin V Apoptosis Detection Kit (BD Pharmingen, Cat#556547) according to the manufacturer's protocol. Cell apoptosis results were analyzed using FlowJo software (v10). The gating strategy is provided in Supplementary Fig. 9. For cell viability assay, cells were seeded at a density of $5 \times 10^3$ cells per well in triplicate in 24-well plates. The next day, cells were treated with various reagents as indicated for 3 days. Cells for glucose deprivation were seeded and cultured in the glucose-sufficient medium at a density of $5 \times 10^4$ cells per well in a 24-well plate, and the next day, the cells were cultured in the culture medium without glucose for another day. Cell viability was determined at absorbance 450 nm using Cell Counting Kit-8 (CCK-8).

## Statistics and reproducibility

All data were presented as mean ± SD. Statistical parameters of the analyzes are reported in the figure legends. All presented data were analyzed using GraphPad Prism 9 software. All in vitro studies were done three times independently and in vivo studies were done two times independently. Western blot analysis was conducted at least three independent times with similar results. The results for the 4NQO mouse model of HNSCC were the pool of two independent experiments. A two-tailed Student's t-test was performed between two groups, and a one-way analysis of variance (ANOVA) with post hoc Bonferroni's test was used for multiple comparisons. Two-way ANOVA with Bonferroni correction was used for subcutaneous xenograft size analysis. Two levels of significance were determined: $*p < 0.05$, $**p < 0.01$, with NS indicating no significance. TCGA data analyzes are from TIMER2.0[60] and cBioPortal[61,62]. No statistical method was used to predetermine the sample size. The sample sizes were determined based on our previous studies and the experience of the authors. No data were excluded from the analyzes. The Investigators were not blinded to allocation during experiments and outcome assessment.

## Reporting summary

Further information on research design is available in the Nature Portfolio Reporting Summary linked to this article.

## Data availability

The raw RNA-seq, ChIP-seq, and whole exome sequencing (WES) data have been deposited at the Gene Expression Omnibus (GEO) under the accession number GEO: GSE237454, GSE234640, and GSE234824, respectively. All processed data are available within the Article, the Supplementary Information, and the Source Data file. Source data are provided with this paper.

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

## Acknowledgements

We thank Dr. Zhong Chen (NIH) and Dr. Thomas E. Carey (University of Michigan) for providing SCC46 and SCC74A cell lines. This work was supported by NIH/NIDCR grants R01DE029173, R01DE030445 and R01CA236878.

## Author contributions

W.L., H.C., and C.Y.W. designed the experiments and wrote the manuscript. W.L. and H.C. performed most of the experiments and data

analyzes. J.W. and W.L. analyzed RNA-seq data and ChIP-seq data. B.H., P.Z., and X.Y.L. assisted with animal studies, and A.E. and W.C. performed some histological staining. S.K. and A..J. generated and provided KMT2D-mutated PDX. All authors read and approved the final manuscript.

## Competing interests

W.L., H.C., and C.Y.W. are listed as inventors on intellectual property owned by UCLA related to targeting KMT2D-mutant tumors. CYW received research funding unrelated to this project from RNAimmune Inc, is a co-founder and equity holder in WZJ Therapeutics and has served as a compensated consultant for Curigin Inc and Acroimmune Inc, all unrelated to the current work. AJ has stock/options ownership in Suvica and Champions Oncology; AJ institution has contracts with Cantargia, DebioPharm, Genentech, Iovance, Khar Biopharma, Merck, Moderna, Pfizer, Sanofi, and SQZ for clinical trials where AJ is the local PI. C.Y.W., W.L., and H.C. have filed a patent application for the findings reported in the manuscript. The remaining authors declare no competing interests.
