## [Peer Review File · Nature Communications]

Histone-methyltransferase KMT2D Deficiency Impairs the Fanconi Anemia/BRCA Pathway upon Glycolytic Inhibition in Squamous Cell CarcinomaReviewers' Comments:

Reviewer #1:

Remarks to the Author:

The manuscript by Liu and colleagues describes an unexpected link between loss of functional FA pathway throughout KMT2D (MLL4) deficiency and glycolysis in HNSCC. Specifically, the authors show that KMT2D deficiency promotes HNSCC invasiveness and metastasis through increased glycolysis. KMT2D deficiency seems to impair the activation of the FA pathway only under glycolytic inhibition, which renders cells sensitive to DNA damaging agents and PARP inhibitors. Moreover, the authors show that KMT2D deficiency reprograms the epigenetic landscape in HNSCC and impairs transcription of some FA genes. The authors conclude that HNSCC epigenetic reprogramming upon KMT2D loss impairs the expression of the FA pathway thus sensitizing cells to DNA damaging drugs upon glycolytic inhibition. Overall, the findings described in the MS about the role of KMT2D deficiency combined with glucose deprivation are of potential interest for the tumorigenesis fields, and the link to the dysregulation of the FA DNA repair pathway has potential implications relevant for the FA field. The author's conclusions are supported by the experimental approaches and the experiments are scientifically sounded. Some experimental approaches such as the PDX and orthotopic mouse models are neat and demonstrates that glucose deprivation in the absence of KMT2D sensitizes HNSCC to a variety of DNA damaging agents. However, However, the data presented in the manuscript suggesting that KMT2D phenotypes upon glucose deprivation are mediated by the FA pathway-mediated mitochondrial stress are less clear, and some author's statements are not entirely supported by the experimental approaches described in this manuscript, which need to be addressed for publication in Nature Communications.

Major points:

1-. The authors title "KMT2D mutation drives HNSCC hypersensitive to DNA damaging agents by impairing the FA/BRCA Pathway upon glycolytic inhibition" is an overstatement and overinterpretation. The authors nicely show that loss of KMT2D increases glycolysis, promotes aggressive HNSCC growth and downregulates expression of some FA genes. If their hypothesis held true, then loss of function of the FA genes should show similar phenotypes to KMT2D depletion, meaning that KMT2D and FA loss are epistatic toward the described phenotypes. Moreover, the authors arise other explanations for their results as the implication of DNA repair pathways other than FA/BRCA, which it makes more sense. The authors could address this by:

- i) Further validations are required to address that KMT2D loss of function compromises the FA/BRCA pathway upon glycolytic inhibition. The authors should determine FA/BRCA protein levels in SCC23 WT and KO cells by other means than qPCR (Western blot or IF).
- ii) Generation of SCC23 cell lines lacking any of the FA genes shown to be downregulated in the RNA-seq experiment (FANCB;L;D2 or I) and test cellular phenotypes (annexin V, caspase 3, sensitivity to 2-DG, MMC, olaparib or their combination), and the tumorigenic phenotype on the orthotopic xenographs (as described in 4b) in nude mice.
- iii) If the FA pathway is epistatic to KMT2D on the described phenotypes upon 2-DG treatment, then cells lacking both cellular pathways (FA/BRCA and KMT2D pathways) should behave as one of the single mutant cells for the observed phenotypes upon glycolytic shutdown. The authors should generate KMT2D^{-/-} SCC23 cell lines lacking any FA/BRCA genes (as described before, FANCB^{-/-}; L^{-/-}; D2^{-/-} or I^{-/-}) and perform epistasis analysis on the described cellular and tumor phenotypes upon glucose deprivation.

2-. To address the FA pathway disfunction upon glucose deprivation in the KMT2D deficient cells, the authors monitored FANCD2 foci formation upon 2-DG, MMC or combined treatment and found no foci formation. D2 foci formation occurs only in S phase of the cell cycle. This reviewer wonders if 2-DG treatment affects cell cycle stages of KMT2D deficient cells, as one alternative explanation for the results would be that glucose deprivation would arrest/blocks cell cycle of KMT2D deficient cells. The

authors should perform the same assay including an S-phase marker (BrdU) to examine if cell cycle stages are altered under these conditions. FACS analysis would also be informative.

3-. Related to figure 3 g, h, i,: i) The authors examined several phenotypes such as accumulation of ROS, mitochondrial fragmentation and mitochondrial membrane potential, which are normally associated to FA deficiency. However, most of these cellular phenotypes are not FA specific, they rather indicate cellular stress. Therefore, most of these assays may correlate not only with deficiency in many DNA repair pathways other than FA. I suggest removing these data from the MS. ii) Also related to figure 3 g, h, i,: There are some cellular phenotypes which are hallmarks of FA, such as chromosome breakages and radial formation. Did the authors evaluate, as a FA deficiency readout, the frequency of chromosome aberrations and radial formation under their glucose deprivation conditions in KMT2D deficient cells?

4-. Related to figure 4 g, h, i,: The authors claim that Kmt2d HT mice exposed to 4-NQO followed by 2DG + MMC presented smaller tumor areas compared to single treatment controls. To correlate tumor formation with dysfunctional FA pathway upon 2-DG treatment, it would be interesting to correlate FA protein levels (FancD2 or FancI expression by IHC) to ascertain tumor regression by 3-DG with absence of the FA pathway

Minor points:

- Some RNA-seq target genes (figure must be validated by western blot, to demonstrate that downregulation of mRNA levels impact on protein levels. It would good to include some non-affected genes as controls.

- There are few typos throughout the text. Please change accordingly.

Reviewer #2:

Remarks to the Author:

The authors wisely planned drug combinations as treatment for HNSCC based on the downstream effects of the KMT2D mutation, which is the most mutated epigenetic modifier in this cancer type. Although the authors performed an analysis of the TCGA database showing a high frequency of KMT2D mutation in cases of head and neck SCC (16%), regardless of HPV infection, it would be important to show the relevance of this mutation in the clinical scenario, showing the genetic mutation status and its impact on overall survival, disease-free survival or correlation with TNM stage. The use of 2DG plus Olaparib or cisplatin has never been previously tested in mouse models. The results of the manuscript are innovative, with translational potential, as such combinations have proven to be effective in vivo, including with the use of PDX models. The experiments were very well planned, conducted, and consistent with the methods. However, specifically in the PDX trials, the authors should inform in the text that these experiments were carried out using subcutaneous tumors, otherwise the analysis of metastatic potential could be requested as previously analyzed in the orthotopic model. Furthermore, a brief comment should be added about this limitation in the discussion section. In the summary session, the authors forgot to add "under glycolysis inhibition" or "under glucose deprivation" at the end of the sentence "Mechanistically, loss of KMT2D reprograms epigenomic landscapes in HNSCC and impairs expression of FA genes by transiting its promoter /enhancer states to inactive states." Overall, the manuscript is very well written, well presented, and provides substantial mechanistic insights into HNSCC drug resistance using epigenomic flaws to properly target cancer cells. A potential future experiment to elevate the quality of this study would be the use of spatial transcriptomics to further provide a comprehensive understanding of the tumor microenvironment and its changes in response to the KMT2D mutation, although this approach may be expensive and time-consuming.

Reviewer #3:

Remarks to the Author:

The manuscript by Liu et al. investigates the impact of prevalent KMT2D mutations in HNSCC and explores potential therapeutic opportunities. The authors demonstrate that KMT2D-deficient HNSCC cells exhibit elevated expression of glycolysis genes, correlating with aggressive tumor growth. Glucose deprivation or inhibiting glycolysis using 2-DG induced significant cell death in KMT2D null cells. The author further showed that the Fanconi Anemia (FA) pathway was down regulated after glycolytic inhibition only in KMT2D null cells. FA pathway induces DNA damage repair and suppresses tumor progression. The differential regulation of FA pathway genes by KMT2D under glucose deprivation or 2-DG treatment led to oxidative stress and cell death. These results point to the beneficial combinatorial treatment of 2-DG and PARP inhibitors. Finally, the authors showed that combining 2-DG with chemotherapy or PARP inhibitors induced tumor growth inhibition in KMT2D KO cells. Overall, the manuscript presents compelling findings. The finding that KMT2D KO affects FA pathway genes in a metabolic state dependent manner is novel and interesting. However, several major concerns should be addressed:

1. Although the directed regulation of FA pathway genes by KMT2D is well-demonstrated, the manuscript lacks clarity on why this regulation is glycolysis dependent. It remains unclear what drives increase of KMT2D occupancy at FA genes in Glc- cells.
2. The mechanism by which KMT2D affects glycolysis genes is not elucidated. It is important to clarify whether these regulations are direct or indirect. Are changes in glycolysis more of reflection of aggressive nature of KMT2D mutant cells?
3. To strengthen the paper, direct demonstration of changes in metabolites associated with glycolysis should be included alongside gene expression changes.
4. Figures 5a and 5b are confusing and may suggest their removal from the main figures or manuscript. The global changes observed could be reflective of growth status rather than being KMT2D-dependent. It is advisable to focus the analyses on KMT2D direct targets. Since most changes in KMT2D/H3K4me1 occurred at enhancers, it is surprising changes at FA genes were mostly at gene promoters. Were there any changes at distal enhancers for FA genes?

Minor comments:

1. Supplementary Figure 2a and 2d were unclear. How do they indicate double KO of KMT2D?
2. What is the molecular weight of KMT2D in Supplementary Figure 2e? The sigma antibody only detects 100kDa band.
3. Why one clone of KMT2D KO show increased KMT2D expression in Fig S2b? What is the explanation? Might need to show protein levels for other KMT2s since transcript levels do not reflect KO status.
4. Text need to be further edited for clarity, e.g.,
Pg 4. this set of patients ----- this patient cohort
Pg 5. truncations (60.7%), which are known to cause loss-of-function of genes
Pg 13-14. Transcription inhibition should be 'transcription repression'

We would like to express our sincere gratitude to the reviewers for their meticulous reading of our manuscript and their invaluable comments. Their insights have significantly contributed to the enhancement of our work.

Reviewer #1

Major points:

1. The authors title “KMT2D mutation drives HNSCC hypersensitive to DNA damaging agents by impairing the FA/BRCA Pathway upon glycolytic inhibition” is an overstatement and overinterpretation.

Based on Reviewer 1's suggestion, we have changed the title of our manuscript to “KMT2D Deficiency Impairs FA/BRCA Pathway upon Glycolytic Inhibition in Squamous Cell Carcinoma”.

i) Further validations are required to address that KMT2D loss of function compromises the FA/BRCA pathway upon glycolytic inhibition. The authors should determine FA/BRCA protein levels in SCC23 WT and KO cells by other means than qPCR (Western blot or IF).

We examined the protein levels of FANCD2 and FANCI in SCC23 WT and KO cells by Western blot. Consistent with the downregulated mRNA levels, Western blot analysis confirmed that the protein levels of FANCD2 and FANCI were also decreased in KMT2D-KO SCC23 cells compared with KMT2D-WT SCC23 cells after glucose deprivation (**Fig. 3f**).

ii) Generation of SCC23 cell lines lacking any of the FA genes shown to be downregulated in the RNA-seq experiment (FANCB;L;D2 or I) and test cellular phenotypes (annexin V, caspase 3, sensitivity to 2-DG, MMC, olaparib or their combination), and the tumorigenic phenotype on the orthotopic xenografts (as described in 4b) in nude mice.

As Reviewer 1 suggested, we generated FANCD2 knockdown (FANCD2-KD) SCC23 cells using lentiviral shRNA targeting FANCD2 (**Supplementary Fig. 7b**) and then subjected them to the treatments. There was modestly increased apoptosis in FANCD2-KD SCC23 cells compared with FANCD2-KD SCC23 cells by the treatment of 2-DG, MMC, or olaparib alone. However, the combination treatment of 2-DG plus MMC or 2-DG plus olaparib significantly induced more pronounced cell apoptosis in FANCD2-KD SCC23 cells compared with FANCD2-WT SCC23 cells (**Supplementary Fig. 7c**). Western blot showed that 2-DG plus MMC (**Supplementary Fig. 7d**) or 2-DG plus olaparib (**Supplementary Fig. 7e**) potentially induced caspase-3 activation in FANCD2-KD SCC23 cells. Next, we injected FANCD2-KD SCC23 cells into the tongue of nude mice. While the treatment with 2-DG, MMC, or olaparib alone modestly inhibited orthotopic tumor growth of FANCD2-KD SCC23 cells, the combination of 2-DG with MMC or olaparib drastically inhibited the tumor growth of FANCD2-KD SCC23 cells (**Supplementary Fig. 7f**).

iii) If the FA pathway is epistatic to KMT2D on the described phenotypes upon 2-DG treatment, then cells lacking both cellular pathways (FA/BRCA and KMT2D pathways) should behave as one of the single mutant cells for the observed phenotypes upon glycolytic shutdown. The authors should generate KMT2D-/SCC23 cell lines lacking any FA/BRCA genes (as described before, FANCB-/-; L-/-; D2-/- or I-/-) and perform epistasis analysis on the described cellular and tumor phenotypes upon glucose deprivation.

As Reviewer 1 suggested, we knocked down FANCD2 in KMT2D-KO SCC23 cells (FANCD2-KD/KMT2D-KO SCC23) (**Supplementary Fig. 7g**). Upon glucose deprivation, we did not observe any significant differences in apoptosis determined by annexin V (**Supplementary Fig. 7h**) and cleaved Caspase-3 (**Supplementary Fig. 7i**) between FANCD2-KD/KMT2D-KO SCC23 cells and the control KMT2D-KO SCC23 cells. Moreover, 2-DG had a similar inhibitory effect on tumor growth of FANCD2-KD/KMT2D-KO SCC23 cells and KMT2D-KO SCC23 cells in vivo (**Supplementary Fig. 7j**). Taken together, these results indicate that the FA pathway is epistatic to KMT2D upon glucose deprivation or 2-DG treatment.

2-. To address the FA pathway dysfunction upon glucose deprivation in the KMT2D deficient cells, the authors monitored FANCD2 foci formation upon 2-DG, MMC, or combined treatment and found no foci formation. D2 foci formation occurs only in the S phase of the cell cycle. This reviewer wonders if 2-DG treatment affects cell cycle stages of KMT2D deficient cells, as one alternative explanation for the results would be that glucose deprivation would arrest/blocks cell cycle of KMT2D deficient cells. The authors should perform the same assay including an S-phase marker (BrdU) to examine if cell cycle stages are altered under these conditions. FACS analysis would also be informative.

Reviewer 1 made an interesting and important point. Indeed, it is possible that KMT2D-deficient cells block FANCD2 foci formation due to cell cycle arrest under glucose deprivation. GO analysis revealed that the downregulated genes by the loss of KMT2D were enriched in DNA replication under glucose deprivation (**Fig. 3b**). We performed BrdU incorporation and propidium iodide staining assays as Reviewer 1 suggested. Indeed, KMT2D loss significantly decreased BrdU incorporation in 2-DG treated KMT2D-KO SCC23 cells compared with KMT2D-WT SCC23 cells (**Supplementary Fig. 4e**). Furthermore, KMT2D-KO SCC23 cells exhibited a significant decrease in the S-phase of cell cycle compared with KMT2D-WT SCC23 cells upon 2-DG treatment by flow cytometry analysis (**Supplementary Fig. 4f**). Therefore, cell cycle arrest caused by KMT2D loss upon 2-DG treatment further contributed to the dysfunction of the FA pathway.

3-. Related to figure 3 g, h, i,: i) The authors examined several phenotypes such as accumulation of ROS, mitochondrial fragmentation and mitochondrial membrane potential, which are normally associated to FA deficiency. However, most of these cellular phenotypes are not FA specific, they rather indicate cellular stress. Therefore, most of these assays may correlate not only with deficiency in many DNA repair pathways other than FA. I suggest removing these data from the MS. ii) Also related to figure 3 g, h, i,: There are some cellular phenotypes which are hallmarks of FA, such as chromosome breakages and radial formation. Did the

authors evaluate, as an FA deficiency readout, the frequency of chromosome aberrations and radial formation under their glucose deprivation conditions in KMT2D deficient cells?

Indeed, we cannot exclude cellular stress and other DNA repair pathways other than the FA pathway that affects ROS and mitochondria under glucose deprivation or 2-DG treatment. As Reviewer 1 suggested, we removed these data from our manuscript. To accurately evaluate the FA pathway, we examined MMC-induced chromosome breaks and radials in KMT2D-WT and KMT2D-KO SCC23 cells. The frequency of chromosome breaks and radial formations induced by MMC were much higher in KMT2D-KO SCC23 cells than in KMT2D-WT cells under 2-DG plus MMC treatment (**Fig. 3g-i**), indicating that KMT2D loss leads to the FA pathway impairment under glycolytic inhibition.

4-. Related to figure 4 g, h, i,: The authors claim that Kmt2d HT mice exposed to 4-NQO followed by 2DG + MMC presented smaller tumor areas compared to single treatment controls. To correlate tumor formation with dysfunctional FA pathway upon 2-DG treatment, it would be interesting to correlate FA protein levels (FancD2 or FancI expression by IHC) to ascertain tumor regression by 3-DG with absence of the FA pathway.

As Reviewer 1 suggested, we did the IHC staining for FANCD2 in 4NQO-induced *Kmt2d*-HT mouse HNSCC. The results showed that the protein levels of FANCD2 were decreased after 2-DG or 2-DG plus MMC treatment (**Supplementary Fig. 5c**), which is associated with the tumor regression under 2-DG or 2-DG plus MMC treatment in *Kmt2d*-HT mouse HNSCC (**Fig. 4g**).

Minor points:

Some RNA-seq target genes (figure must be validated by western blot, to demonstrate that downregulation of mRNA levels impact on protein levels. It would good to include some non-affected genes as controls.

We added Western blot results for FANCD2 and FANCI (**Fig. 3f**), which were consistent with the downregulated mRNA levels in KMT2D-KO SCC23 cells under glucose deprivation. We also showed the protein levels of non-affected FA genes, such as FANCG and FANCL, in KMT2D-KO and KMT2D-WT SCC23 cells under glucose deprivation. (**Supplementary Fig. 4a**).

There are few typos throughout the text. Please change accordingly.

We have corrected the typos throughout the text.

Reviewer #2:

1. Although the authors performed an analysis of the TCGA database showing a high frequency of KMT2D mutation in cases of head and neck SCC (16%), regardless of HPV infection, it would be important to show the relevance of this mutation in the clinical scenario, showing the genetic mutation status and its impact on overall survival, disease-free survival or correlation with TNM stage.

We downloaded the HNSCC TCGA data and stratified the patients into two groups: the KMT2D mutated group with loss-of-function mutations and the unaltered KMT2D wildtype group. Then, we did the Kaplan-Meier curve analysis of the overall survival (OS) of these two groups. The results showed that loss-of-function mutation of KMT2D was significantly associated with poorer overall survival (OS) in HNSCC (**Supplementary Fig. 1d**).

2. However, specifically in the PDX trials, the authors should inform in the text that these experiments were carried out using subcutaneous tumors, otherwise the analysis of metastatic potential could be requested as previously analyzed in the orthotopic model. Furthermore, a brief comment should be added about this limitation in the discussion section.

We specifically state that the PDX experiments were carried out using subcutaneous tumors (on page 18) which could not examine the metastatic potentials.

3. In the summary session, the authors forgot to add “under glycolysis inhibition” or “under glucose deprivation” at the end of the sentence “Mechanistically, loss of KMT2D reprograms epigenomic landscapes in HNSCC and impairs expression of FA genes by transiting its promoter /enhancer states to inactive states.”

We have revised our summary based on our new findings. The “glycolysis inhibition” has been incorporated into the sentence “Mechanistically, while glucose deprivation facilitates KMT2D occupancy to the promoter/enhancer of FA genes by activating AMP-activated protein kinase, KMT2D loss leads to reprogramming epigenomic landscapes in HNSCC and impairs the expression of FA genes by transiting their promoter/enhancer states from active to inactive states.” as you suggested (on page 2).

4. A potential future experiment to elevate the quality of this study would be the use of spatial transcriptomics to further provide a comprehensive understanding of the tumor microenvironment and its changes in response to the KMT2D mutation, although this approach may be expensive and time-consuming.

As Reviewer 2 mentioned, this is an expensive and time-consuming experiment. We are unable to get results within a short period. After our project gets funded, we plan to perform this experiment in the near future. We added the comments about KMT2D and tumor microenvironment in the Discussion section (on page 20).

Reviewer #3:

1. Although the directed regulation of FA pathway genes by KMT2D is well-demonstrated, the manuscript lacks clarity on why this regulation is glycolysis dependent. It remains unclear what drives increase of KMT2D occupancy at FA genes in Glc- cells.

This is a very challenging question. It is reported that glucose deprivation activates AMPK signaling to regulate cancer cell growth and survival¹. We observed that AMPK was activated in SCC23 cells upon glucose deprivation or 2-DG treatment (**Supplementary Fig. 6g,h**). We hypothesized that AMPK might be involved in the KMT2D occupancy at FA genes in SCC23 cells under glucose deprivation or 2-DG treatment. We used the specific AMPK inhibitor Compound C to inhibit AMPK activation induced by 2-DG (**Fig. 6i**). Consistent with our hypothesis, ChIP-qPCR results showed that the 2-DG-induced KMT2D occupancies at the promoters of *ATR*, *FANCM*, *REV3L*, and *TOP3A* were significantly blocked by Compound C (**Fig. 6j-m**). As a result, the mRNA levels of *ATR*, *FANCM*, *REV3L*, and *TOP3A* were dramatically decreased by 2-DG and Compound C combination treatment in KMT2D-WT SCC23 cells (**Supplementary Fig. 6i**). Furthermore, cell viability was also greatly decreased upon the combination treatment of 2-DG and Compound C (**Supplementary Fig. 6j**). These results indicate that the activation of AMPK by glycolytic inhibition promotes KMT2D occupancies at the promoters of the FA genes, thereby increasing the expression of FA genes. Because of KMT2D loss, glycolytic inhibition could not induce the FA genes in KMT2D mutant SCC cells. Taken together, these mechanistic findings further enhance the novelty of our manuscript. Of note, KMT2D is a huge protein that is difficult to work with. Currently, we don't know whether AMPK directly phosphorylates KMT2D, thereby promoting KMT2D binding to the promoters of the FA genes. Definitely, it is our future study.

2. The mechanism by which KMT2D affects glycolysis genes is not elucidated. It is important to clarify whether these regulations are direct or indirect. Are changes in glycolysis more of reflection of aggressive nature of KMT2D mutant cells?

KMT2D is a transcriptional co-activator, and its loss will lead to the downregulation of its direct target genes. Since the glycolytic genes were increased upon KMT2D loss (**Fig. 2h-j**), and the *KMT2D* mRNA levels and the mRNA levels of several glycolytic genes, including *ADH5*, *ENO1*, *GAPDH*, *LDHB*, *PDHA1*, and *TPI1*, were negatively correlated by TCGA analysis (**Supplementary Fig. 2i**), it suggests that KMT2D indirectly regulates glycolytic genes. Previously, elegant studies have shown that KMT2D loss indirectly increased glycolysis in lung cancer and melanoma due to the inhibition of *PER2* or *IGFBP5*, which is the transcriptional repressor of glycolytic genes^{2,3}. Although we did not find that KMT2D loss in HNSCC affected the expression of *PER2* or *IGFBP5*, it is most likely that KMT2D mutation utilizes a similar mechanism to promote glycolytic gene expression. In the future, it is important to identify potential transcriptional repressors of glycolytic genes in HNSCC regulated by KMT2D. Moreover, since it is already known that KMT2D mutation indirectly regulates glycolytic genes in

several human cancers, we did not focus our work on this aspect in our manuscript. Nevertheless, we added more discussions in the Discussion section (page 20).

Our in vivo data suggest that increased glycolysis might promote the tumor growth of KMT2D mutant cells. To investigate whether KMT2D-mutant HNSCC manifested the aggressive behavior of cells through increased glycolysis, we performed a transwell invasion assay. To avoid 2-DG-induced cell apoptosis in KMT2D-KO SCC23 cells, we treated the cells along with KMT2D-WT SCC23 cells with or without 2-DG only for 16 hours. The transwell assay showed increased invasiveness of KMT2D-KO SCC23 cells compared with KMT2D-WT cells. However, 2-DG treatment almost eliminated the enhanced invasion by KMT2D loss (**Supplementary Fig. 3o**), indicating that enhanced glycolysis contributes to the aggressive nature of KMT2D-deficient HNSCC.

3. To strengthen the paper, direct demonstration of changes in metabolites associated with glycolysis should be included alongside gene expression changes.

In addition to measuring glucose and lactate in the culture medium (**Fig. 2k,l**), we also checked two intracellular glycolytic metabolites, 2-phosphoglycerate (2-PG) and pyruvate. Our results showed that intracellular levels of 2-PG and pyruvate were significantly increased in KMT2D-KO SCC23 cells compared with KMT2D-WT cells (**Fig. 2m,n**).

4. Figures 5a and 5b are confusing and may suggest their removal from the main figures or manuscript. The global changes observed could be reflective of growth status rather than being KMT2D-dependent. It is advisable to focus the analyses on KMT2D direct targets. Since most changes in KMT2D/H3K4me1 occurred at enhancers, it is surprising changes at FA genes were mostly at gene promoters. Were there any changes at distal enhancers for FA genes?

As Reviewer #3 suggested, we moved Figures 5a and 5b to **Supplementary Fig. 6a,b**. Thank you for your suggestion to analyze distal enhancers for FA genes. We defined enhancers by high levels of H3K4me1 and H3K27ac. We checked the distal regions of *ATR*, *FANCM*, *REV3L*, and *TOP3A*, and found that strong peaks of KMT2D, H3K4me1 and H3K27ac were also found at the distal enhancer regions of *FANCM* and *REV3L* (**Supplementary Fig. 6c,d**). Moreover, the levels of H3K4me1 and H3K27ac were significantly decreased at the enhancer regions of *FANCM* and *REV3L* in KMT2D-KO cells upon glucose deprivation (**Supplementary Fig. 6c,d**), indicating that KMT2D loss reduces enhancer activities to downregulate the expression of the FA genes in HNSCC cells upon glucose deprivation.

Minor comments:

1. Supplementary Figure 2a and 2d were unclear. How do they indicate double KO of KMT2D?

Sorry for the confusion. The results from Sanger sequencing shown in **Supplementary Fig. 2a and 2e** were used to verify CRISPR/Cas9-mediated editing of *KMT2D* around the PAM sequence. These two figures showed CRISPR/Cas9-mediated *KMT2D*-edited cells have mismatched bases around the

PAM sequence, which indicates that there are indeed mutations in at least one allele and the other allele is either wild type or has different mutations. The results from our Sanger sequencing could not indicate a double knockout of *KMT2D*. It is only used to check the editing of *KMT2D*. Western blot was subsequently used to confirm *KMT2D* double knockout, which showed that no *KMT2D* proteins had been detected in the *KMT2D*-KO cells (**Fig. 2a** and **Supplementary Fig. 2f**).

2. What is the molecular weight of *KMT2D* in Supplementary Figure 2e? The sigma antibody only detects 100kDa band.

KMT2D has 5,537 amino acids with a molecular weight of approximately 600 kDa. The largest protein marker (Thermo Fisher Scientific, Cat#LC5699) we used for Western blot is 460 kDa, and the band of *KMT2D* shown in **Fig. 2a** and **Supplementary Fig. 2f** is above 460 kDa. Regarding the *KMT2D* antibodies from Sigma, Sigma used a Myc-tagged truncated *KMT2D*, amino acids 2916-3785 fragment (~120 kDa) for Western blot to show that their antibodies can detect recombinant *KMT2D* fragments. This is why the Sigma antibody only detected a band of around 100 kDa.

3. Why one clone of *KMT2D* KO show increased *KMT2D* expression in Fig S2b? What is the explanation? Might need to show protein levels for other *KMT2s* since transcript levels do not reflect KO status.

We did not observe significant changes in *KMT2D* mRNA levels between *KMT2D*-KO and *KMT2D*-WT SCC23 cells. We repeated the RT-PCR of *KMT2D* mRNA and showed the results in **Supplementary Fig. 2b**. We also examined the protein levels of *KMT2A*, *KMT2B*, and *KMT2C*, and their protein levels were not changed after *KMT2D* depletion in SCC23 cells (**Supplementary Fig. 2c**).

4. Text need to be further edited for clarity, e.g., Pg 4. this set of patients ----- this patient cohort. Pg 5. truncations (60.7%), which are known to cause loss-of-function of genes. Pg 13-14. Transcription inhibition should be 'transcription repression'

We have corrected these issues and further edited our text as suggested.

References

1. Jeon SM, Chandel NS, Hay N. AMPK regulates NADPH homeostasis to promote tumour cell survival during energy stress. *Nature* **485**, 661-665 (2012).
2. Alam H, *et al.* KMT2D Deficiency Impairs Super-Enhancers to Confer a Glycolytic Vulnerability in Lung Cancer. *Cancer Cell* **37**, 599-617 e597 (2020).
3. Maitituoheti M, *et al.* Enhancer Reprogramming Confers Dependence on Glycolysis and IGF Signaling in KMT2D Mutant Melanoma. *Cell Rep* **33**, 108293 (2020).

Reviewers' Comments:

Reviewer #1:

Remarks to the Author:

This reviewer would like to acknowledge the authors for their effort and novel experimental data that clarifies the role of KMT2D (MLL4) in maintaining the FA pathway active during glucose deprivation and I would like to congratulate them for the work providing a novel layer of regulation of the FA pathway in HNSCC in vivo. However, regarding the cell cycle alterations upon 2-DG treatment described by the authors, there is still a possibility that most of the observations pointing to downregulation of the FA pathway under glycolytic inhibition by 2-DG in the absence of KMT2D could be just a mere consequence of the activation of the p53/p21 axis-mediated G1 arrest leading to a marked decreased of S phase cells, and therefore lack of activation of the FA pathway.

This reviewer suggests introducing several changes:

- subheading in page 9: To clarify that upon glycolytic inhibition there is a alterations in cell cycle populations leading to diminish S phase cells (cells where the FA pathway is mainly active) "KMT2D deficiency causes cell cycle arrest and FA pathway impairment upon glycolytic inhibition" instead of "KMT2D deficiency causes FA pathway impairment upon glycolytic inhibition".
- Page 10: in the chromosome breakage analysis, in figure 3e, the authors should show all the experimental conditions, not only MMC+2DG, to ascertain that the FA defects observed is a consequence of the combined glycolytic inhibition and KMT2D.

Reviewer #2:

Remarks to the Author:

The authors have adequately addressed my comments in the revised version of the manuscript. Therefore, I have no further comments.

Reviewer #3:

Remarks to the Author:

The authors have satisfactorily addressed all my previous concerns. The new metabolite analyses and the epigenomic/enhancer studies greatly strengthened the main conclusions. It will be of great interests to the broad audience of Nature Communication.

Reviewer #1

- 1) - subheading in page 9: To clarify that upon glycolytic inhibition there is a alterations in cell cycle populations leading to diminish S phase cells (cells where the FA pathway is mainly active) “KMT2D deficiency causes cell cycle arrest and FA pathway impairment upon glycolytic inhibition” instead of “KMT2D deficiency causes FA pathway impairment upon glycolytic inhibition”.

We sincerely thank Reviewer #1's positive comments and suggestions. Based on Reviewer #1's suggestion, we have changed the subheading on page 9 to “KMT2D deficiency causes cell cycle arrest and FA pathway impairment upon glycolytic inhibition”.

- 2) - Page 10: in the chromosome breakage analysis, in figure 3e, the authors should show all the experimental conditions, not only MMC+2DG, to ascertain that the FA defects observed is a consequence of the combined glycolytic inhibition and KMT2D.

In our pilot study, we examined chromosome breakage under conditions of 2-DG treatment alone and we found that nearly 100% of cells, both KMT2D-WT and KMT2D-KO cells, showed no detectable DNA breaks. Therefore, it is not necessary to show these results. Since increased chromosomal breakage and radial formation after exposure to DNA cross-linking agent MMC are diagnostic features of Fanconi anemia (FA) cells, we focused on investigating FA pathway impairment in KMT2D-deficient cells under combined MMC and 2-DG treatment conditions. Therefore, we analyzed chromosome breakage exclusively under MMC plus 2-DG treated conditions in KMT2D-WT and KMT2D-KO cells (Fig. 3g).

Reviewer #2

The authors have adequately addressed my comments in the revised version of the manuscript. Therefore, I have no further comments.

We deeply appreciate Reviewer #2's careful evaluation of our manuscript.

Reviewer #3 (Remarks to the Author):

The authors have satisfactorily addressed all my previous concerns. The new metabolite analyses and the epigenomic/enhancer studies greatly strengthened the main conclusions. It will be of great interests to the broad audience of Nature Communication.

We deeply appreciate Reviewer #3's careful evaluation of our manuscript.